# Application of the neuropeptide NPVF to enhance angiogenesis and osteogenesis in bone regeneration

Hongping Yu[1,6], Yanyi Wang[2,3,6], Junjie Gao [4,5], Youshui Gao [4✉], Chao Zhong [2,3✉] & Yixuan Chen [4✉]

The brain-bone regulatory system regulates skeletal homeostasis via bioactive neuropeptides, yet the underlying mechanism remains elusive. Here, we report the role of the neuropeptide VF (NPVF, VPNLPQRF-NH$_2$) in enhancing both angiogenesis and osteogenesis in a rat skeletal system and the potential pathways involved. An in vitro study revealed that NPVF not only promotes migration and angiogenesis of human umbilical vein endothelial cells (HUVECs) by activating NPFFR1, which leads to upregulation of miR-181c-3p and downregulation of Argonaute1 (AGO1), but also mediates osteogenic differentiation of bone mesenchymal stem cells (BMSCs) via the Wnt/β-catenin signaling pathway. To improve the stability and bioavailability and thus efficacy of NPVF as a promoter of in vivo bone regeneration, we genetically engineered amyloid-NPVF-fusion proteins and utilized them as self-assembling nanofiber coatings to treat bone defects in a rat calvarial defect model. We found that a porous hydroxyapatite scaffold loaded with the NPVF peptide-fused amyloid coating substantially enhanced angiogenesis and site-specific fresh bone in-growth when implanted in calvarial defects. Taken together, our work uncovered a previously undefined crosstalk between the brain and bone by unveiling the role of NPVF in bone tissue and demonstrated a viable method for promoting bone tissue repairs based upon self-assembling NPVF-containing protein coatings.

[1] Department of Orthopedic Surgery, The First Affiliated Hospital of Xiamen University, School of Medicine, Xiamen University, Xiamen, Fujian 361005, China. [2] Center for Materials Synthetic Biology, Shenzhen Institute of Synthetic Biology, Shenzhen Institutes of Advanced Technology, Chinese Academy of Sciences, Shenzhen 518055, China. [3] CAS Key Laboratory of Quantitative Engineering Biology, Shenzhen Institute of Synthetic Biology, Shenzhen Institutes of Advanced Technology, Chinese Academy of Sciences, Shenzhen 518055, China. [4] Department of Orthopedic Surgery, Shanghai Sixth People's Hospital Affiliated to Shanghai Jiao Tong University School of Medicine, Shanghai 200233, China. [5] Ningbo Institute of Life and Health Industry, University of Chinese Academy of Science, Ningbo, Zhejiang, China. [6]These authors contributed equally: Hongping Yu, Yanyi Wang. ✉email: gaoyoushui@sjtu.edu.cn; chao.zhong@siat.ac.cn; yixuanchen_sjtu@163.com

Bone homeostasis involves bone formation by osteoblasts and bone destruction by osteoclasts; these processes are interconnected and tightly regulated, assuring the maintenance of skeletal health[1]. Bone homeostasis has been found to be closely associated with the brain-bone regulatory system, primarily through various bioactive neuropeptides, such as substance P (SP), neuropeptide Y (NPY), and calcitonin gene-related peptide (CGRP)[2–4]. Recent studies have revealed that these neuropeptides are closely involved in osteogenesis and angiogenesis[2,3,5,6], two hallmark events in bone formation[7,8].

The neuropeptide FF (NPFF) system, first identified in bovine brain extract in 1985[9,10], has been found to be involved in a variety of physiological processes, including food intake, blood pressure regulation, memory, insulin release, neural regeneration, metabolic disease, and cardiovascular activity[11–16], via activation of a $G_{i/o}$ protein-coupled receptor located in the nervous system[17,18]. Despite these advances, no previous studies have reported a connection between the NPFF system and bone homeostasis. However, as an increasing number of neuropeptides have been recently discovered to participate in the brain-bone regulatory system, the tantalizing possibility has been raised that the NPFF system may have a largely ignored yet important physiological role in the skeletal system.

In this work, using NPVF (a representative of the NPFF family, it is named NPVF after the abbreviation for its first and last amino acid.) as a model system, we probed the possible roles of the NPFF family in both osteogenesis and angiogenesis during bone formation. We revealed that NPVF could indeed promote both osteogenesis and angiogenesis in a rat skeletal system (Fig. 1a). Building on these discoveries, we next explored the potential application of the NPVF peptide for in vivo bone regeneration. Because the application of peptide drugs is often hampered by their structural vulnerability in serum and rapid clearance via renal filtration[19], we genetically engineered NPVF-amyloid protein fusions and utilized them as self-assembling nanofiber coatings to treat bone defects in a rat calvarial defect model (Fig. 1b). We found that implantation of a porous HAp scaffold containing NPVF peptide-fused amyloid coatings could significantly enhance angiogenesis and fresh site-specific bone ingrowth in calvarial bone. In short, our studies clearly revealed a crosstalk between the brain and bone by unveiling the exact molecular role of NPVF in bone tissue formation, and demonstrated the feasiblilty of a cell-free and growth factor-free method for efficient treatment of bone repairs based upon self-assembling NPVF-containing protein coatings.

## Results

### NPVF induces HUVEC migration and angiogenesis by activating NPFFR1.
Inspired by recent discoveries that neuropeptide systems might be involved in vascular development and angiogenesis in bone homeostasis[5,6,20], we set out to assess whether the NPVF peptide, a well-known representative of the NPFF neuropeptide family, participates in bone homeostasis. Note that the NPFF system exerts a variety of physiological functions via selective activation of the $G_{i/o}$ protein-coupled receptors Neuropeptide FF Receptor 1 (NPFFR1) and NPFFR2[21,22], and NPVF is a highly selective ligand of NPFFR1[21]. Therefore, we first investigated the expression level of the *NPFF1R* in HUVECs by qPCR. The average Ct value for *NPFFR1* was $19.20 \pm 0.22$, indicating that NPFFR1 was highly expressed in HUVECs (Fig. 2a). Next, we explored the role of NPFFR1 in HUVECs using the selective agonist NPVF. A Cell Counting Kit-8 (CCK-8) assay showed that low dose of NPVF (0.01, 0.1 and 1 nM) had no toxic effect on the proliferation of HUVECs, while high dose of NPVF (10 nM) had a slight inhibition on the proliferation of HUVECs (Fig. 2b).

Hence, low doses of NPVF (0.01, 0.1, and 1 nM) were chosen for subsequent experiments. We next studied the effect of NPVF on the migration capacity of HUVECs by performing wound healing (36 h) and transwell migration (24 h) assays. The wound healing area of HUVECs in the NPVF group was much larger than that in the control group, and the NPFFR1 antagonist RF9 significantly counteracted the healing-promoting effect of NPVF (Fig. 2c-d). The results of the transwell assay also revealed that RF9 could neutralize the NPVF-induced migration of HUVECs (Fig. 2e-f). These results together indicated that NPVF could promote HUVEC migration and proliferation. Next, we explored the effect of NPVF on angiogenesis using a tube formation assay (6 h). As shown in Fig. 2g-i, the NPVF group had more loop structures and a higher number of branch points than the control group, while RF9 reversed the NPVF-induced promotion of HUVEC tube formation. Finally, qPCR and enzyme-linked immunosorbent assay (ELISA) were employed to detect the expression of angiogenesis-associated genes and proteins, respectively, in HUVECs. The qPCR results revealed that the expression of *VEGF*, *EGF*, and *PDGF* in HUVECs remarkably increased after NPVF treatment, and the addition of RF9 could offset this promotion effect. (Figs. 2j-2l). ELISA analysis of the protein expression level of VEGF yielded the same conclusion. (Fig. 2m). In short, our experimental results suggested that NPVF significantly promotes migration and angiogenesis by activating NPFFR1 in HUVECs.

### NPVF-induced migration and angiogenesis of HUVECs is mediated by MiR-181c-3p.
MicroRNAs (miRNAs) are a family of evolutionarily conserved endogenous gene repressors present in all living organisms. They are extensively involved in the posttranscriptional regulation of gene expression in various biological contexts, including the systemic control of many biological processes, such as osteogenesis and angiogenesis[23–26]. For example, miR-497/195 regulates bone homeostasis by regulating the coupling of angiogenesis and osteogenesis[27], and miR-136-3p regulates vascularization by targeting PTEN[28].

Blood vessel invasion plays a vital role in new bone regeneration by providing oxygen, nutrients, cells, and hormones[29–31]. Hence, to further explore the molecular mechanism underlying NPVF promotion of angiogenesis of HUVECs, we performed miRNA sequencing (miRNA-seq) of HUVECs after NPVF treatment (48 h). The miRNA-seq results showed that six miRNAs were upregulated and thirteen miRNAs were downregulated in the NPVF-treated HUVECs compared with untreated HUVECs (Fig. 3a). Among all the miRNAs with upregulated expression, we were particularly interested in miR-181c, the target gene of which plays a substantial role in endothelial cell function[32,33]. Thus, we further investigated its expression level by qPCR. Our results showed that the expression level of miR-181c-3p in HUVECs was remarkably upregulated under NPVF induction (Fig. 3b). We further validated the effect of miR-181c-3p on HUVECs by performing a series of experiments, namely a wound healing experiment (36 h), transwell assay (24 h), tube formation assay (6 h), qPCR, and ELISA analysis, where cells were treated with miR-181c-3p or NC-mimics (as negative control). The wound healing experiment and transwell assay both demonstrated that HUVECs in the miR-181c-3p group exhibited better migration and proliferation capacities than those in the control group (Fig. 3c-f). In addition, quantitative analysis of tube formation revealed that the miR-181c-3p group formed more loop structures and branch points, suggesting higher angiogenic activity (Fig. 3g-h, Fig. S1a). Moreover, the qPCR and ELISA (48 h) results indicated that expression of angiogenesis-associated markers (VEGF, EGF,

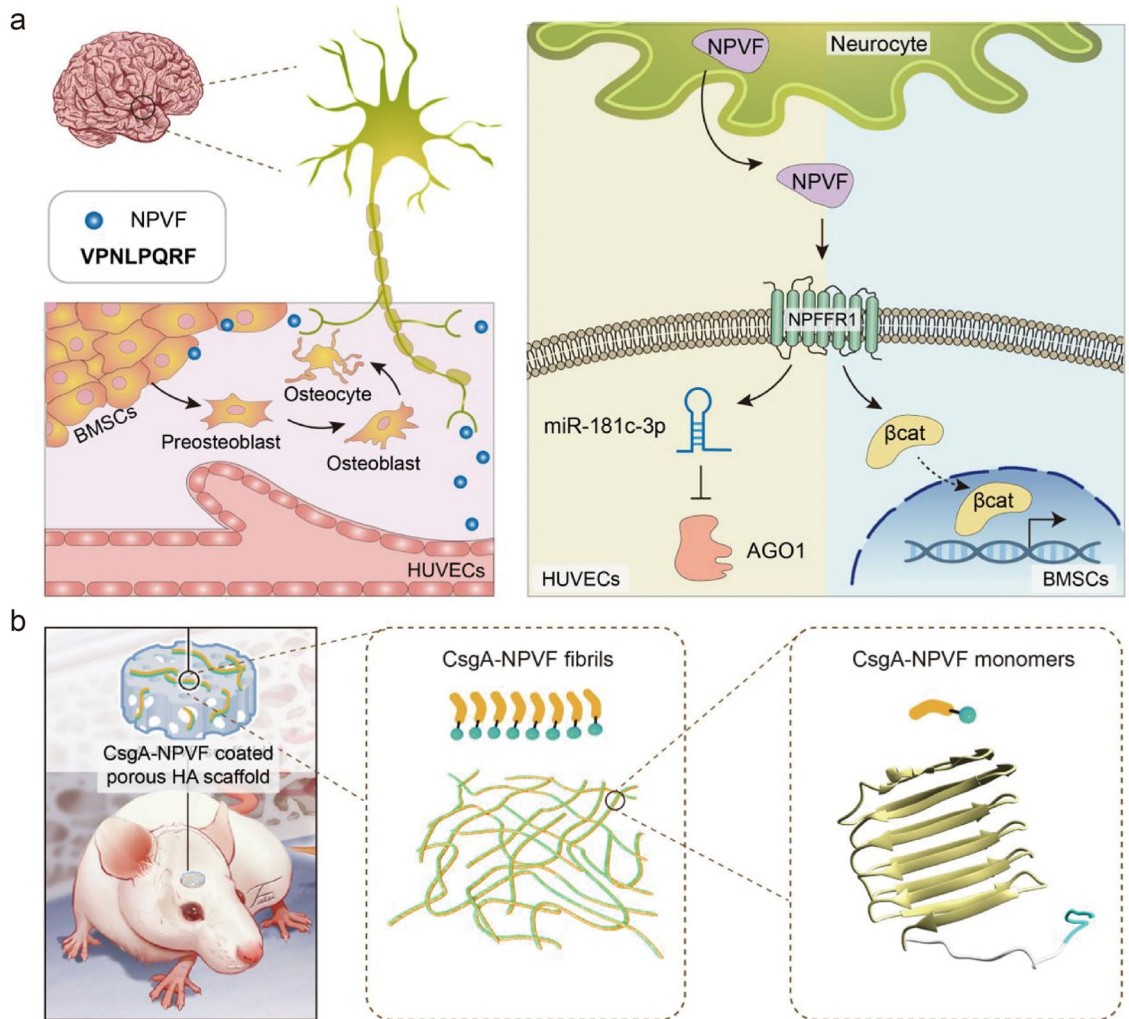

**Fig. 1 Unveiling the role and mechanism of NPVF in mediating crosstalk between the brain and bone and application of self-assembling NPVF-containing protein coatings for in vivo bone defect repair. a** Schematic image depicting the promoting effect of the neuropeptide NPVF on both angiogenesis and osteogenesis. NPVF is an eight-amino acid neuropeptide secreted by the human hypothalamus, and it regulates various physiological processes including bone homeostasis maintenance by activating its $G_{i/o}$ protein-coupled receptor located in the nervous system (left panel). The amino acid sequence of NPVF is presented in the upper box. The right panel shows the signaling pathway by which NPVF promotes osteogenic differentiation of BMSCs and vascularization of HUVECs. NPVF binds preferentially to NPFFR1 in the central nervous system, promotes angiogenesis of HUVECs via the miR-181c-3p/AGO1 pathway, and mediates osteogenic differentiation of BMSCs via the Wnt/β-catenin signaling pathway. **b** Illustration of bone fracture repair through the implantation of a CsgA-NPVF nanofiber-coated porous hydroxyapatite scaffold in a rat calvarial defect model. The CsgA amyloid can self-assemble into nanofibers to form a robust proteinaceous coating on the surface of given substrate. Coupling the coat-forming property of CsgA with NPVF induced new bone formation. Displaying the engineered CsgA-NPVF functional nanofibers on the HAp scaffold substantially facilitated bone regeneration. The right panel shows the simulated structure of the CsgA-NPVF monomer revealed by molecular dynamics simulations.

PDGF, and FGF) was significantly increased in HUVECs after miR-181c-3p treatment (Fig. S1b-f). Based on the above results, we inferred that miR-181c-3p mediates the NPVF-induced migration and angiogenesis in HUVECs.

MiRNAs generally function by base pairing with the 3'-untranslated regions (3'-UTRs), thereby downregulating the post-transcriptional expression of mRNAs. Bioinformatic analysis of miR-181c-3p showed that it might target the 3'-UTR sequence of AGO1 (Fig. 3i). The wild-type and mutant AGO1 3'-UTR sequences were thus cloned into the psiCHECK2 luciferase vector to explore whether miR-181c-3p directly regulates AGO1 production. Transfection of miR-181c-3p mimic significantly inhibited the luciferase activity of the wild-type 3'-UTR group. In contrast, the luciferase activity of the mutant 3'-UTR group was unaffected by the miR-181c-3p mimic (Fig. 3j). Moreover,

transfection of the miR-181c-3p mimic distinctly decreased the expression level of AGO1 in HEK293T cells, as compared with those transfected with the miR-NC (Normal Control) mimic (Fig. 3k).

We further investigated the relationship between AGO1 and NPVF in HUVECs. NPVF treatment substantially reduced the AGO1 gene and protein expression levels in HUVECs (Fig. S1g-h). Next, we probed the exact role of AGO1 in the signaling pathway in HUVECs by knocking down AGO1 using small interfering RNA (siRNA). A western blotting assay revealed that the control siRNA (NC siRNA) did not affect the protein expression level of AGO1 in HUVECs, while AGO1 siRNA distinctly downregulated it (Fig. 3l-m). Moreover, AGO1 siRNA distinctly promoted the migration and tube formation of HUVECs, which mimicked the effect of NPVF (Fig. 3n-p, Fig. S1i-k).

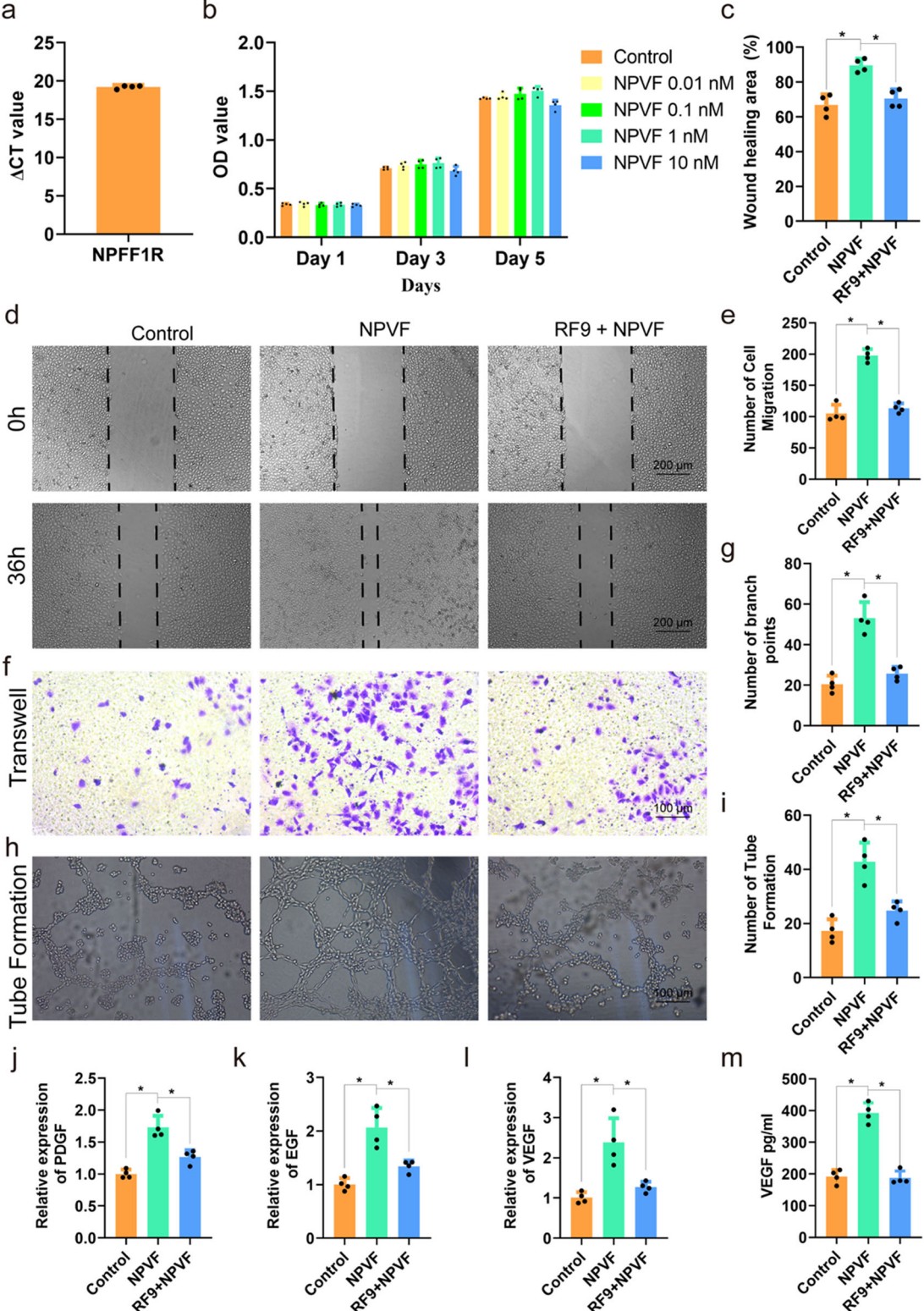

Collectively, these results indicated that NPVF regulates cell migration and angiogenesis by activating NPFFR1, which then mediates the miR-181c-3p/AGO1 pathway.

**The WNT/β-catenin signaling pathway mediates the NPVF-induced osteogenic differentiation of BMSCs.** We next explored whether the NPVF peptide mediates osteogenesis, another hallmark

event of bone regeneration. Indeed, preliminary studies revealed that NPVF promoted the osteogenic differentiation of BMSCs in a dose-dependent manner (Fig. S2). To probe the underlying mechanism, we carried out transcriptome analysis of BMSCs in the presence and absence of NPVF. Heatmap and KEGG pathway enrichment analysis indicated that the WNT pathway was highly enriched in the presence of NPVF (Fig. 4a-b), implying that the WNT pathway might play a key role in the NPVF-induced

**Fig. 2 NPVF promotes the migration and tube formation of HUVECs. a** The expression level of NPFFR1 in HUVECs. Results are presented as means ± S.E.M. of four independent experiments. **b** The proliferation of HUVECs treated with NPVF was evaluated by the CCK-8 assay. Results are presented as means ± S.E.M of four independent experiments. **c, d** Wound healing of HUVECs (36 h). Results are presented as means ± S.E.M of four independent experiments. The $P$ values are 0.0005 (left) and 0.0106 (right). $^*P < 0.05$, one-way ANOVA. **e, f** Transwell migration of HUVECs (24 h). Results are presented as means ± S.E.M. of four independent experiments. The $P$ values are 0.0004 (left) and 0.001 (right). $^*P < 0.05$, one-way ANOVA. **g–i** Tube formation by HUVECs (6 h). Results are presented as means ± S.E.M of four independent experiments. $P$ values from left to right are 0.0003, 0.0007, 0.0008, and 0.0038. $^*P < 0.05$, one-way ANOVA. **j–l** The gene expression levels of *PDGF*, *EGF*, and *VEGF* in HUVECs in the presence or absence of NPVF ± RF9 as determined by qPCR. Results are presented as means ± S.E.M of four independent experiments, and the $P$ values from left to right are 0.0002, 0.0044, 0.0015, 0.009, 0.0042, and 0.0111. $^*P < 0.05$, one-way ANOVA. **m** The protein expression level of VEGF in HUVECs in the presence or absence of NPVF ± RF9. Results are presented as means ± S.E.M of four independent experiments, and the $P$ values are 0.0004 (left) and 0.0004 (right). $^*P < 0.05$, one-way ANOVA.

osteogenic differentiation of BMSCs. A western blotting assay revealed that NPVF could distinctly increase the level of β-catenin in BMSCs, while the NPFFR1 antagonist RF9 and the WNT antagonist JW74 both abolished the upregulation of β-catenin in NPVF-treated BMSCs (Fig. 4c). Alizarin Red S (ARS, 21 days) and alkaline phosphatase (ALP, 7 days) assays were then carried out to explore the role of the Wnt/β-catenin pathway in NPVF-induced osteogenic differentiation of BMSCs. Both the staining and corresponding quantitative results indicated that NPVF remarkably promoted extracellular mineralization and ALP activity in BMSCs, while these effects were abolished in the presence of either RF9 or JW74 (Fig. 4d, Fig. S3). In conclusion, these results revealed that NPVF promotes osteogenic differentiation of BMSCs via the Wnt/β-catenin signaling pathway.

**CsgA-NPVF scaffolds enhance bone regeneration in vivo**. After verifying that NPVF can induce angiogenesis and osteogenesis of HUVECs, we further explored the application potential of NPVF in bone defect repair. We chose to construct a functional recombinant protein, CsgA-NPVF, by fusing the N-terminus of NPVF with the amyloid protein CsgA, a major protein component of *Escherichia coli* curli biofilm, which is well known for its universal ability to form coatings around a given substrate through amyloid self-assembly (Fig. 5a). This strategy offers a facile way to display functional peptides on amyloid scaffolds and overcomes the problems of serum instability and low bioavailability of peptide drugs in preclinical application. After induction of bacterial expression and protein purification, we demonstrated by Transmission Electron Microscopy (TEM) that NPVF-containing nanofibers could only be formed when expressed in fusion with CsgA (Fig. S4).

We performed Congo red staining (Fig. 5b) and a Thioflavin T assay (Fig. 5c) to verify the amyloid features of both CsgA and CsgA-NPVF; their fibrous structures were further confirmed by atomic force microscopy (AFM) imaging (Fig. 5d). Quartz crystal microbalance (QCM) tests indicated strong adsorption of the CsgA-NPVF nanofibers to the HAp substrate (Fig. 5e). After overnight immersion in fresh purified protein solution, we observed that the CsgA-NPVF protein could form robust nanofiber coatings on the surface and internal interfaces of a porous 3D HAp scaffold. Morphological changes recorded by scanning electron microscopy (SEM) imaging validated the successful formation of CsgA-NPVF nanofiber coatings on the scaffold surfaces (Fig. 5f). To further explore the stability of the recombinant protein coating, we performed in vitro protein release experiments using simulated body fluid (SBF). Both uncoated scaffolds and CsgA-NPVF-coated scaffolds were immersed in SBF solution (pH=7.4) and incubated at 37 °C for two months. His-tag labelled CsgA-NPVF proteins released into the SBF solution was detected using a commercially available enzyme-linked immunosorbent assay (ELISA) kit. During the 60-day test, the amount of supernatant protein in the CsgA-NPVF

groups was almost the same as that in the control group (Fig. S5), indicating that the CsgA-NPVF coatings had good stability.

Owing to its good biocompatibility and surface adhesion capacity, CsgA has been used in various medical applications, such as surface modification of cell culture scaffolds and bone implants[34], secretion of CsgA-fused cytokine for the treatment of inflammatory bowel disease (IBD)[35]. Based on previous studies, we reasonably speculated that fusion with CsgA could improve the stability of NPVF peptides while preserving their biological activity. Before animal experiments, we tested the cytocompatibility of different scaffolds using BMSCs and found that compared with cells grown on the empty HAp scaffolds and CsgA-coated scaffolds, cells grown on the CsgA-NPVF-coated scaffolds had the best adhesion capacity and the highest viability (Fig. S6).

We next evaluated the efficacy of the NPVF-containing protein-coated HAp scaffolds in assisting bone regeneration in vivo using a standard rat calvarial defect model. We first set out to investigate the expression level of NPFFR1 in rat calvarias by immunohistochemistry (IHC). The results indicated that NPFFR1 was highly expressed in rat calvarias (Fig. S7), which was consistent with the results from the in vitro experiment (Fig. 2a). Furthermore, at 12 weeks after transplantation, micro-CT reconstruction data indicated that more fresh bone formation formed into the direct 3D space of the CsgA-NPVF protein-coated scaffolds in the defects than in those treated with the control groups (bare HAp scaffolds and CsgA protein-coated scaffolds) (Fig. 6a). Quantitative analysis revealed that both the bone mineral density (BMD, 1.538 g/cm³ VS 1.276 g/cm³) and the bone volume/tissue volume (BV/TV, 69.069% VS 50.142%) ratio in the CsgA-NPVF group were higher than those in the control groups (Fig. 6b-c). Hematoxylin & eosin (H&E) staining and Masson staining all demonstrated enhanced osteoblastic marker expression and bone matrix formation occurred in the 3D space of the CsgA-NPVF protein-coated scaffolds in the defects than in those treated with the control groups (bare HAp scaffolds and CsgA protein-coated scaffolds) (Fig. 6d-e). Moreover, IHC analysis revealed much higher expression levels of CD31 and COL I in the bone matrix formation for the CsgA-NPVF group (Fig. 6f-g). It should be noted that HAp scaffolds are a well-recognized and standard biomaterial used as bone graft substitutes, providing an objective evaluation of bone regeneration[36–39]. The HAp-CsgA-NPVF group showed significantly more new bone in-growth and neovascularization in the defect area than the pure HAp scaffold group and HAp-CsgA group, suggesting that our cell-free and growth factor-free composite is superior to this traditionally employed scaffold. Overall, the CsgA-NPVF-coated scaffold enhanced bone regeneration via coupling of osteogenesis and angiogenesis in rats.

## Discussion

Neuropeptides are the messengers in the brain-bone axis. They are released from sensory nerve endings and exert distinct

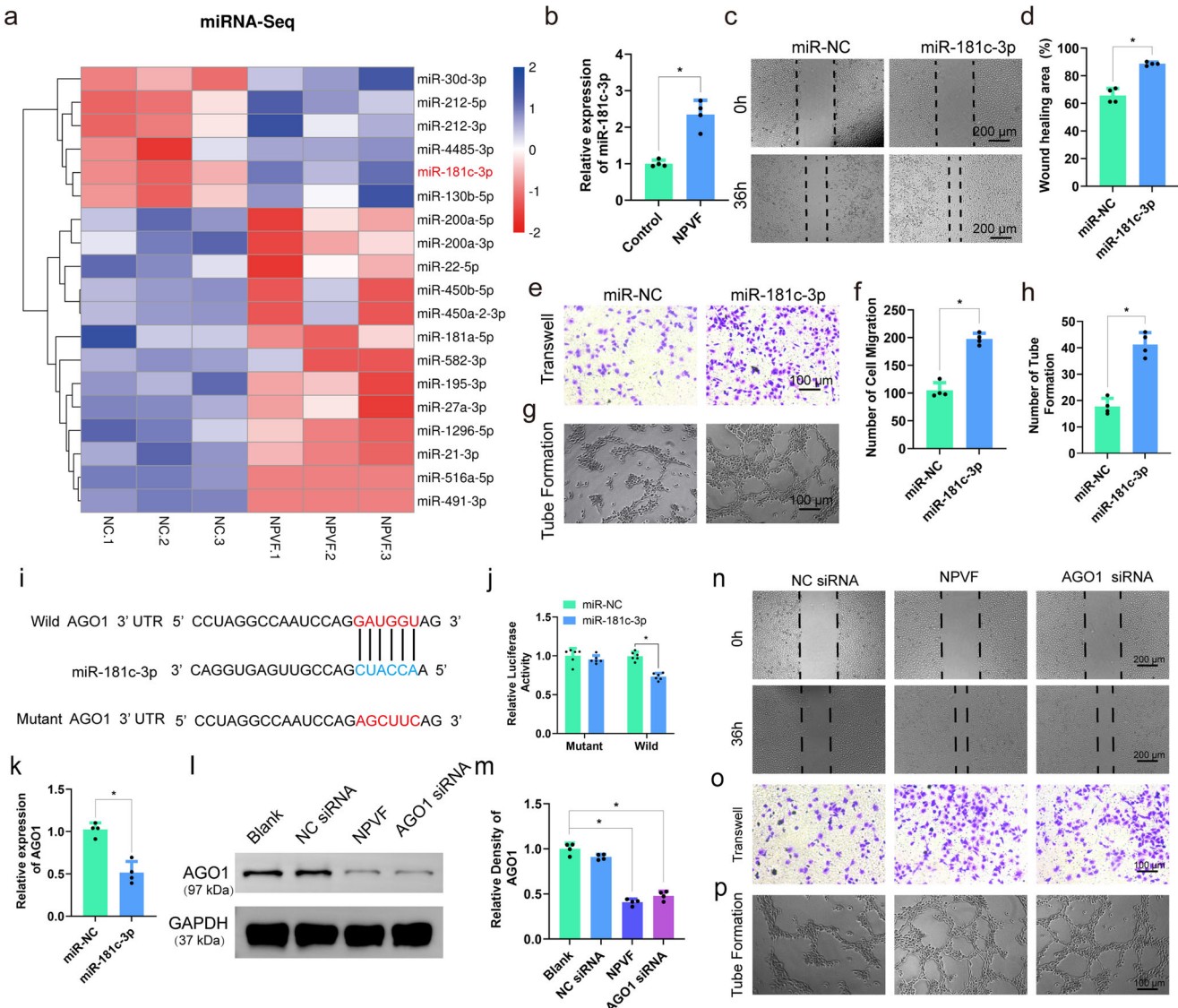

**Fig. 3 miR-181c-3p mediates the NPVF neuropeptide-induced migration and tube formation of HUVECs. a** Heatmap of miRNA sequencing results for HUVECs treated with NPVF (48 h). The color represents the relative expression value of miRNA genes. **b** The gene expression level of *miR-181c-3p* in HUVECs after treatment with NPVF. Results are presented as means ± S.E.M. of four independent experiments, and the *P* value is 0.0005. *$^*P < 0.05$, Student's *t*-test. **c, d** Wound healing of HUVECs (36 h). Results are presented as means ± S.E.M. of four independent experiments, and the *P* value is 0.0004. $^*P < 0.05$, Student's *t*-test. **e, f** Transwell migration of HUVECs (24 h). Results are presented as means ± S.E.M. of four independent experiments, and the *P* value is 0.0001. $^*P < 0.05$, Student's *t*-test. (**g, h**) Tube formation by HUVECs (6 h). Results are presented as means ± S.E.M. of four independent experiments, and the *P* value is 0.0001. $^*P < 0.05$, Student's *t*-test. (**i**) Construction of the wild-type *AGO1* 3'-UTR and mutant *AGO1* 3'-UTR plasmids. The potential target sites of *miR-181c-3p* in the *AGO1* 3'-UTR are shown. **j** Normalized luciferase activity of the vectors after miR-181c-3p overexpression in HEK293T cells. Results are presented as means ± S.E.M. of six independent experiments, and the *P* value is 0.0039. $^*P < 0.05$, Student's *t*-test. **k** The level of *AGO1* was remarkably downregulated after miR-181c-3p overexpression in HEK293T cells. Results are presented as means ± S.E.M. of four independent experiments, and the *P* value is 0.0005. $^*P < 0.05$, Student's *t*-test. (**l, m**) AGO1 siRNA was used to knock down the expression of AGO1 protein in HUVECs. Results are presented as means ± S.E.M. of four independent experiments, and the *P* values are 0.0004 (left) and 0.0001 (right). $^*P < 0.05$, one-way ANOVA. **n** The wound healing of HUVECs. (**o**) Transwell migration of HUVECs. **p** Tube formation by HUVECs.

functions on skeletal system, including BMSC differentiation[40,41], cartilage metabolism[2,4], fracture healing[42,43], and bone-fat equilibrium[20]. Despite advances in elucidating the functions of these neuropeptides, no previous studies have reported a connection between the NPFF system and bone homeostasis. In this study, we demonstrate for the first time that the NPVF peptide (a representative of the NPFF system) significantly promotes both angiogenesis and osteogenesis. In addition, we identified the signaling pathways through which NPVF participates in maintaining bone homeostasis. The finding of a crosstalk between the

brain and bone suggests that other neuropeptides may also play an important role in bone homeostasis and that NPVF could potentially be used as a peptide drug to treat bone fractures. Moreover, the identification of the critical pathways involved in mediating angiogenesis and osteogenesis paves the way for further investigation and exploitation of NPVF in therapeutic applications. Notably, NPVF represents a type of endogenous peptide that exhibits promising therapeutic efficacy. The main advantages of peptides include their safety, the high target affinity, targeting a wide range of molecules, low toxicity and low

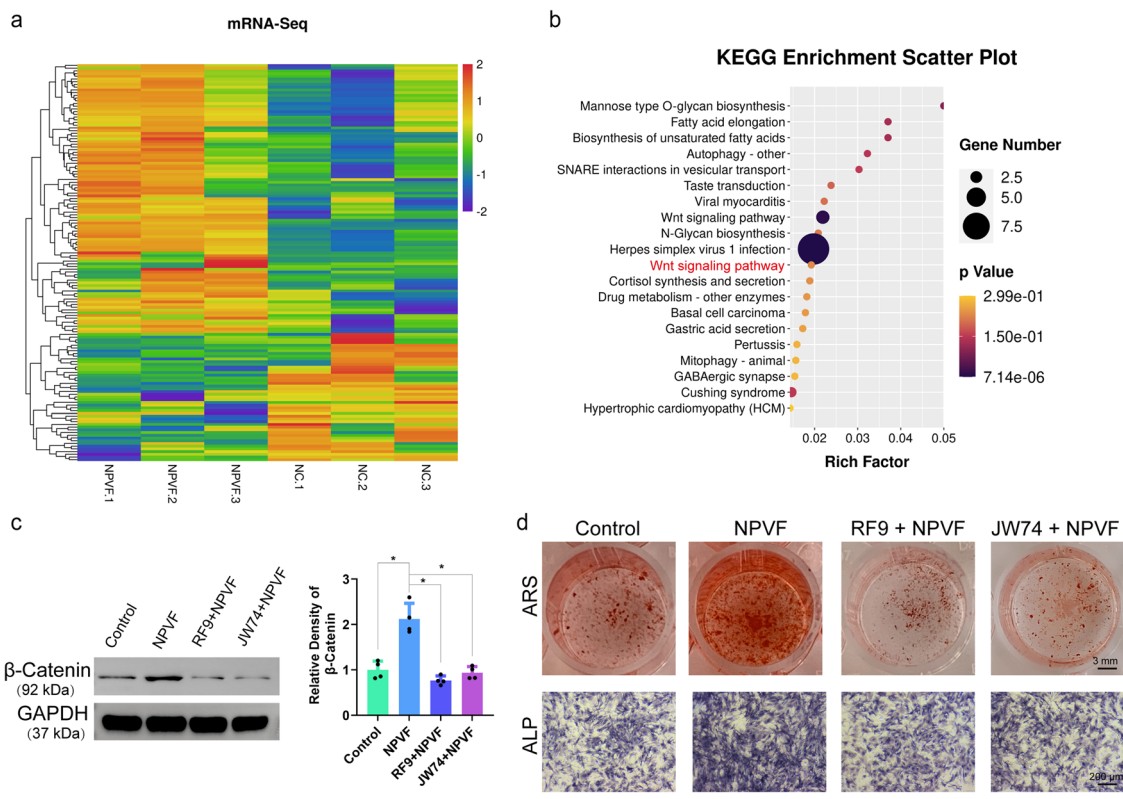

**Fig. 4 The WNT/β-catenin pathway mediates the NPVF-induced osteogenic differentiation of BMSCs. a** The results of miRNA sequencing of BMSCs treated with NPVF. The color represents the relative expression value of miRNA genes. **b** KEGG enrichment analysis of BMSCs after treatment with NPVF. **c** The protein expression of β-catenin in BMSCs. Results are presented as means ± S.E.M. of four independent experiments, and the P values from left to right are 0.0013, 0.0754, and 0.092. *P < 0.05, one-way ANOVA. **d** ARS and ALP staining of BMSCs undergoing osteogenic differentiation.

immunogenicity[44–46]. The functional coatings based on the NPVF peptide-containing fusion proteins might avoid problems including immune rejection and donor-donor variability where existing xenogenous cell-laden tissue engineering strategy frequently meets[47–49].

Beyond the analysis of the molecular regulatory pathway, we explored the application of the NPVF peptide in the pre-clinical treatment of bone defects. Although there is increased interest in peptide therapeutics, the bioavailability of these peptides is usually hampered by their structural vulnerability in the serum environment and rapid clearance via renal filtration[50,51]. To resolve these issues, we took advantage of a programmable self-assembling CsgA nanofiber coating when developing a strategy for the pre-clinical application of the NPVF peptide. CsgA is an amyloid protein that assembles into a fibrous coating on a given substrate in solution. Another distinctive feature of CsgA is that the amyloidogenic core of the assembled structure does not significantly diverge from that of the original amyloid backbone structure when domains with the appropriate size and conformation are present[52]. In our work, we fused the NPVF peptide to the C-terminus of the CsgA subunit to fabricate a porous HAp scaffold with a functional CsgA-NPVF nanofiber coating. It is well recognized that porous HAp scaffolds are highly biocompatible, osteoconductive, and osseointegrative, and they have excellent mechanical properties, which makes them the most commonly used biomaterial in bone tissue repair[53,54], both for scientific research and clinical applications[55–59]. The porous HAp scaffold we selected is a cell culture insert that has already been commercialized in the field of bone regeneration. The experimental results showed that the CsgA-NPVF nanofiber-coated HAp scaffold exhibited drastically increased osteoinductivity and angiogenic-inductivity compared with those of the uncoated

scaffolds in a rat cranial defect model, demonstrating its high pre-clinical application value.

Note that NPAF (three versions, including the NPSF peptide containing 18 amino acid residues, the 8 amino acid short version of NPAF and a 37 amino acid extended version of RFRP-1, had been described in the literature) is also an important family member of NPFF system[21,60]. We initially chose the shorter polypeptide (NPVF) in our study rather than the longer polypeptide (NPSF) because we rationalize that the fusion of the shorter polypeptide onto CsgA would not disrupt the self-assembly of CsgA protein and thus ensure their fiber coating formation abilities. In addition, our series of previous works mainly focused on the osteo-inductivity of NPVF and the in-depth molecular mechanism (manuscript to be submitted), the selected NPVF had already been validated with specific functions and perfectly met the need of current studies. However, it should be noticed that neither NPSF nor NPAF should be ignored or excluded from the future study of NPFF family, where they might also play critical role. It should be noted that the NPVF itself is an important member in the family of neuropeptide. This work aimed at introducing the possibility of utilizing NPVF in the skeletal system which widens the application of neuropeptides. Combined with our previous findings that NPFF activation of NPFF2 receptor stimulates neurite outgrowth in Neuro 2 A cells via activating the ERK signaling pathway[18], we will further work on how this neuropeptide regulates bone metabolism and the feedback of the skeletal system to the brain in the future.

In summary, we have revealed the role and potential mechanism of neuropeptides in regulating bone formation and developed a generalizable and viable strategy based on neuropeptide-containing protein coatings for bone repair. Recalling the role of neuropeptides in regulating angiogenesis, it

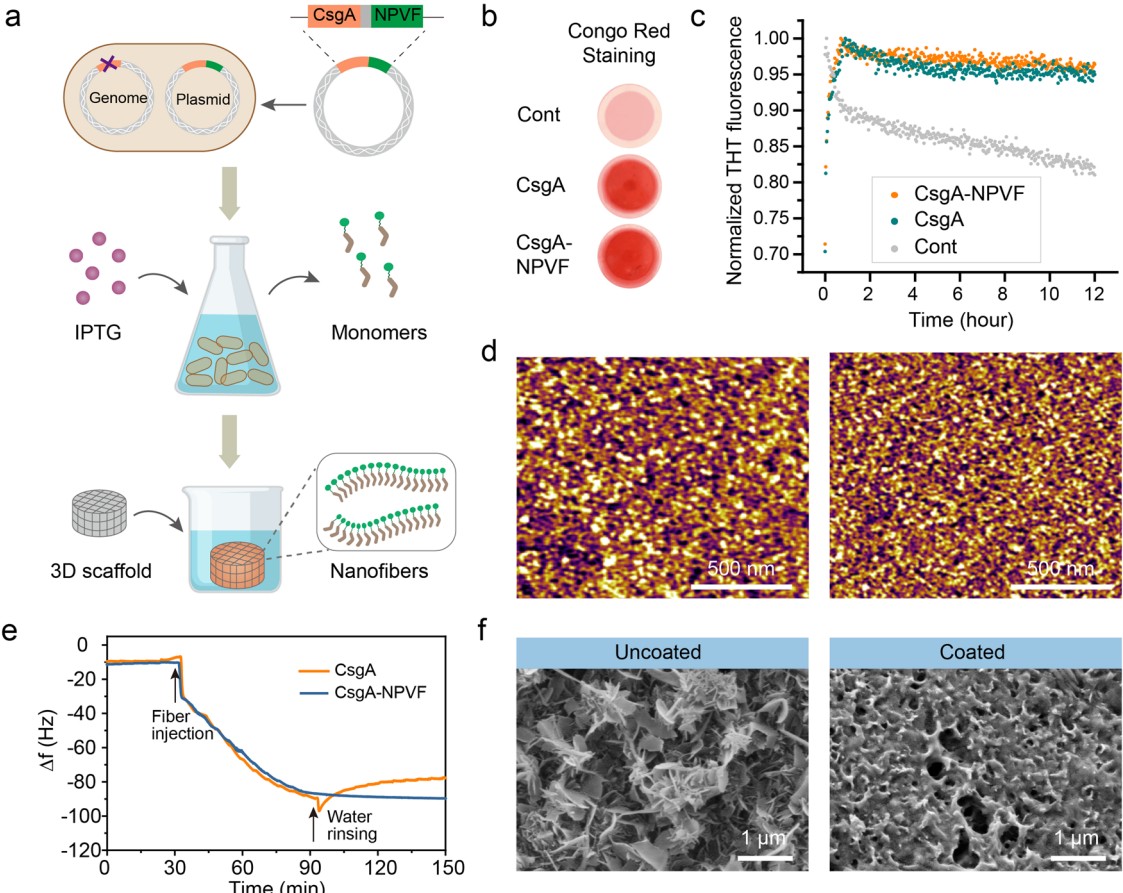

**Fig. 5 Preparation and characterization of the CsgA-NPVF nanofiber coating. a** Schematic diagram depicting the induction of recombinant protein expression by Isopropyl ß-D-1-thiogalactopyranoside (IPTG) and the preparation of nanofiber coatings on a porous 3D HAp scaffold. **b** Congo red staining of the purified CsgA and CsgA-NPVF nanofibers. **c** Assembly kinetics of CsgA and CsgA-NPVF monomer solutions monitored by ThT fluorescence assay. The imidazole buffer used to elute the target proteins during protein purification was used as a control for Congo red staining and ThT fluorescence assay. **d** AFM images showing the morphology of the self-assembled protein nanofibers. **e** Comparison of the adsorption behavior of the CsgA and CsgA-NPVF nanofibers on HAp substrates measured by QCM. **f** SEM images showing the surface morphology of the 3D HAp scaffold before and after the coating process.

is reasonable to speculate that neuropeptides would also provide an effective treatment for various human diseases that are closely related to angiogenesis, such as cancer, obesity, cardiovascular diseases, and chronic inflammation. It is also worth noting that the CsgA nanofiber coating method employed in our work showed great biocompatibility and robustness. The CsgA coating technique alleviates the problem of enzymatic degradation in the physiological environment seen with other frequently used techniques including oral administration and intravenous injection, and it offers long-term protection of the functional peptide, thus substantially enhancing peptide drug bioavailability. Notably, many other CsgA fusion proteins with well-defined biological functions can be precisely engineered using a simple modular genetic design, offering possibilities to functionally modify various graft or scaffold materials in the field of tissue engineering.

## Methods

**Study design**. The objective of this study was to reveal the role and potential mechanism of NPVF in regulating bone formation. We discovered a close relationship between the neuropeptide NPVF and bone formation, and uncovered the underlying molecular mechanisms. Building on these discoveries, we adopted an amyloid protein coating technique to display the NPVF peptide in 3D HAp scaffolds for in vivo treatment of bone defects. HUVECs and BMSCs were used as target cells for the evaluation of NPVF in regulating angiogenesis or osteogenesis, respectively. For the in vitro evaluation of NPVF in regulating angiogenesis, we carried out series of in vitro tests including wound healing assay, transwell assays

and ELISA of VEGF in HUVECs. For revealing of the molecular mechanism miRNA sequencing and dual-luciferase reporter assay were performed. The result indicated the miR-181c-3p was the center mediator which downregulation of AGO1. The evaluation of osteogenesis was performed by PCR, western blots and ARS in BMSCs. Further experiments revealed that NPVF promotes osteogenic differentiation of BMSCs via the Wnt/β-catenin signaling pathway.

To improve the stability and bioavailability and thus efficacy of NPVF as a promoter of bone regeneration, we genetically engineered amyloid-NPVF-fusion proteins and utilized them as self-assembling nanofiber coatings to treat bone defects in a rat calvarial defect model. For the preparation of functional coatings, we constructed a pET22b-*CsgA-NPVF* plasmid and transformed it into the BL21 (DE3) *E. coli* competent cell for recombinant protein purification. The amyloid characteristics of CsgA-NPVF were verified by Congo red staining and Thioflavin T assay, and the fibril morphology was demonstrated by AFM imaging. QCM technique was adopted to reveal the adsorption of proteins on HAp scaffolds. Morphological changes of the scaffold after coating process were observed by SEM imaging. For in vivo studies, animals were randomized to three groups, and no mice were excluded during the experiment. Mice were treated by an operator who was blinded to treatment groups. All tests were replicated by at least three times, and all replicates were successful.

**Cell culture**. The embryonic kidney 293 T cells (HEK93T) and human BMSCs were purchased from the cell bank of the Chinese Academy of Sciences (Shanghai, China) and cultured in Dulbecco's Modified Eagle's Medium (Gibco, Grand Island, NY) and α Minimum Essential Medium (α-MEM, Gibco), respectively, containing 10% fetal bovine serum (FBS) (Invitrogen, Carlsbad, CA) and 1% penicillin/streptomycin (Invitrogen) at 37 ℃ with 5% $CO_2$. HUVECs were purchased from Procell (Wuhan, China) and cultured in endothelial cell medium (ECM, ScienCell, CA). The miRNA sequencing of HUVECs and the mRNA sequencing of BMSCs were performed by LC-bio (Hangzhou, China).

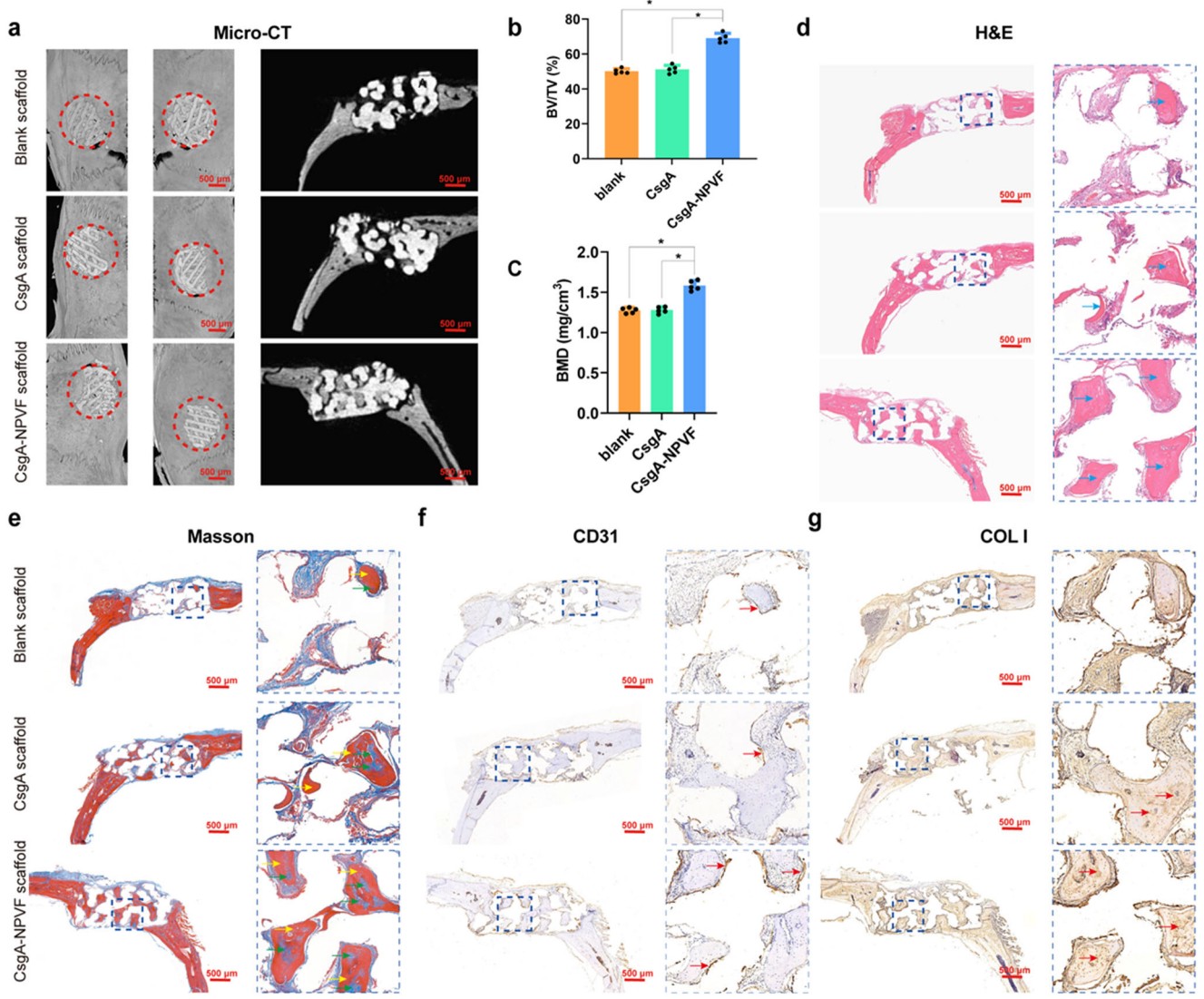

**Fig. 6 CsgA-NPVF nanofiber-coated scaffolds promote bone defect repair in a rat calvarial defect model. a** 3D reconstructed (left) and 2D sagittal (right) micro-CT images of the bone defects. **b, c** The quantification of BMD and BV/TV in the new bone. Results are presented as means ± S.E.M. of five animals, and the *P* values from left to right are 0.0009, 0.0001, 0.0008, and 0.0003. *$P < 0.05$, one-way ANOVA. **d** Histological analysis of new bone in-growth using H&E (blue arrows indicate new bone formation). **e** Masson's histological analysis of new bone formation (green arrows show less mature fresh bone and yellow arrows point to mature bone tissue). **f** Histological observation of CD31 staining (red arrow shows positive staining for CD31). **g** Immunohistochemical analysis of COL I staining (red arrows indicate positive staining for COL I).

**Cell proliferation**. CCK-8 (Beyotime, Jiangsu, China) was used to detect the proliferation of BMSCs and HUVECs after NPVF treatment. In brief, a 96-well plate was seeded with $5×10^3$ cells per well in 100 μL medium at day 0. On days 1, 3, 5 and 7, the medium was refreshed with 10 μL CCK-8 solution and 90 μL medium, then the plate was incubated in an incubator for 2 h. Finally, the absorbance values of the supernatants of each well were detected at 450 nm and the medium was refreshed.

**Angiogenesis and migration of HUVECs**. The wound healing assay was conducted by seeding HUVECs at $5×10^5$ per well in a 6-well culture plate. Then, the confluent cells were scratched using a sterile pipette tip. Cells were then incubated in conditional medium and images of the wounds were captured at 0 h and 36 h. Image J was used to calculate the rates of wound healing. Transwell assays were performed to test the migration of HUVECs after treatment with NPVF, RF9 (Selleck, Shanghai, China), miR-NC mimic, miR-181c-3p mimic, NC siRNA, and AGO1 siRNA (Ruibo, Guangdong, China). In brief, $2×10^4$ cells per well were seeded in the upper chambers of a transwell plate with 200 μL medium and 700 μL of conditional medium was added to the lower chambers. After incubation 24 h at 37 °C, the cells were washed and fixed with 4% paraformaldehyde; cells in the upper surface of the membrane were rubbed away carefully and cells in the lower surface were stained by crystal violet. Finally, the total number of cells in six randomly chosen lower surfaces were counted under a microscope. The tube

formation assay was carried out by adding 50 μL per well Matrigel (BD Bioscience, San Jose, CA) to 96-well plates and allowing gelatinization to proceed at 37°C for 30 min. Then, $4×10^4$ HUVECs/well were seeded on the surface of the Matrigel. A LEICA microscope was used to capture tube formation images 6 h later, and the number of complete capillaries and nodes in each hole was counted.

**Enzyme-linked immunosorbent assay**. HUVECs were incubated in a 6-well plate. After reaching 80% confluence, cells were incubated in serum-free medium for 48 h. Then an ELISA kit (Neobioscience, Shenzhen, China) was used to measure the amount of VEGF in the supernatant. The absorbance values at 450 nm were used to calculate the VEGF content using a standard curve.

**Osteogenesis assay**. To investigate the effect of NPVF, RF9, and JW74 (Selleck) on osteogenic differentiation of BMSCs, cells were incubated in conditional medium for a specific amount of time. In brief, $2×10^5$ cells were seeded in 6-well plates with α-MEM. After cells reached 80% confluence, osteogenic medium (Cyagen, Suzhou, China) was refreshed to induce osteogenic differentiation every two days. ARS staining and ALP staining were performed on days 21 and 14, respectively. A LEICA microscope was used to capture images. The quantification of Alizarin red staining was performed according to the protocol of Hexadecylpyridinium chloride monohydrate (Solarbio, Beijing, China), the OD value was detected at 562 nm by a multifunctional microplate reader after the dissolution of mineralization nodes.

The quantification of ALP staining was performed by an Alkaline phosphatase assay kit (Jiancheng, Nanjing, China) in the manuscript.

**Quantitative real-time polymerase chain reaction**. BMSCs or HUVECs were incubated in 6-well plates with conditional medium for 48 h, and then total miRNA or mRNA was extracted based on the manufacturer's protocols (EZBioscience, USA). Reverse transcriptase reactions consisted of purified RNA and 50 nM RT primer. QPCR was performed on an MX3005P system using an SYBR Premix Ex Taq protocol (EZBioscience). *GAPDH* and *U6* were used as the internal genes for normalization of mRNA and miRNA expression, respectively. The primers are listed in Table S1.

**Western blot analysis**. In brief, BMSCs or HUVECs were lysed with RIPA lysis buffer (1 mM PMSF, Beyotime). The BCA protein assay kit (Beyotime) was used to determine the protein concentrations according to the manufacturer's protocols. An equal amount of protein (35 μg) was electrophoresed on a 10% sodium dodecyl sulfate-polyacrylamide gel, then transferred to a PVDF membrane. The membrane was blocked with 5% (w/v) milk and incubated overnight at 4°C with primary antibodies against GAPDH, AGO1, or β-catenin (1:1000, CST, Shanghai, China). Then, the PVDF membrane was washed and incubated with an HRP-conjugated second antibody (1:5000, CST). Finally, the membrane was washed and reacted with an ECL detection kit (Thermo Scientific, Waltham, MA). Scanning densitometry was used to quantify the signals.

**Dual-luciferase reporter assay**. The wild-type and the mutant fragments of the *AGO1* 3'-UTR sequence including the predicted *miR-181c-3p* target sites (Positions 3932-3938) were directly synthesized (Obio, Shanghai, China) and individually cloned into the 3' end of the psiCHECK2 luciferase vector (Promega; Madison, WI). The Lipofectamine 3000 reagent (Life Technologies) was used to transfect plasmids into the HEK293T cells with the *miR-181c-3p* mimic or miR-NC (Ruibo). A dual-luciferase reporter assay kit (Promega) was used to detect the activities of the Renilla and firefly luciferases at 48 h after transfection. Renilla luciferase activity was normalized to the firefly luciferase activity to calculate the relative luciferase activities.

**Protein purification and coating fabrication**. To create the plasmid for CsgA-NPVF expression, the gene sequence of NPVF was synthesized by GENEWIZ and then inserted into the linearized pET22b-*CsgA* plasmid via one-step Gibson assembly to obtain pET22b-*CsgA-NPVF*. The pET22b-*CsgA* and pET22b-*CsgA-NPVF* plasmids were transformed into BL21 (DE3) *E. coli* competent cells individually. Protein expression and purification were conducted following the protocols described in a previous study[52]. Congo red staining and the ThT assay were performed to characterize the amyloid structure of the proteins following the methods reported in another study[61]. Then the protein solutions were incubated on mica surfaces for 2 hours and washed with deionized water for AFM imaging. Samples were dried under a constant flow of nitrogen gas before imaging. Tapping mode AFM was performed on an Asylum MFP-3D atomic force microscope (Asylum Research) using microcantilevers (k = 26 N/m and ν~ 300 kHz). Next, QCM tests were applied to examine the adsorption capacities of the protein nanofibers on the HAp surface. Nanofiber solutions at an initial concentration of 0.5 mg·ml$^{-1}$ were introduced into HAp-coated QCM chips at a flow rate of 0.02 mL/min using a four-channel Ismatec ICP-N4 peristaltic pump, followed by washing with PBS buffer at the same rate. After incubation in the protein solution overnight, the modified 3D HAp scaffolds were washed with copious amounts of deionized water and dried using a vacuum drying oven, then coated with an 8–10-nm gold layer using a sputter coater for SEM observation. SEM images were taken using a JSM 7800 scanning electron microscope operated at a 5-kV accelerating voltage in secondary electron imaging mode. The 3D HAp scaffolds we used were purchased from INNOTERE company (111CC4), the scaffold consists of synthetic calcium phosphate (mainly α-tricalcium phosphate) and nanocrystalline calcium deficient hydroxyapatite. The scaffolds offer interconnected porosity (3D version), high bioactivity, ease of handling, and high mechanical stability, making them ideal substrates for cell cultures in the field of bone regeneration.

**TEM imaging**. For TEM imaging, a 10 μL droplet of protein was directly deposited on a TEM grid (Zhongjingkeyi Technology, EM Sciences) for 5 min. The excessive solution was wicked away with pieces of filter paper and the samples were rinsed twice with ddH2O by placing 10 μL ddH2O on TEM grid and quickly wicking it off with filter paper. The samples were then negatively stained with 10 μL 2 wt% uranyl acetate for 1 min and dried for 15 min under an infrared lamp. TEM images were obtained on a Tecnai G2 Spirit T12 transmission electron microscope operated at 120 kV accelerating voltage.

**In vitro protein release experiment**. After the coating process, both uncoated HAp scaffolds and CsgA-NPVF-coated scaffolds were immersed in 10 mL SBF solution (pH=7.4) and incubated at 37 °C for two months. His-tag labelled CsgA-NPVF proteins released into the solution were detected using a commercially available His tag ELISA detection kit (Genscript, Nanjing, China). ELISA

experiments were performed according to the instruction at 1, 3, 7, 14, 30 and 60 days after immersion.

**SEM assay and immunofluorescent staining**. SEM assays and immunofluorescent staining were performed to observe the morphology and adhesion of BMSCs on the scaffolds. In brief, after being seed with BMSCs ($1.0 \times 10^4$) in 48-well plates for 48 h, scaffolds were fixed with glutaraldehyde and dehydrated by gradient concentrations of ethanol (30, 40, 50, 60, 70, 80, 90, 100 v/v%). Then, the scaffolds were observed by SEM. For immunofluorescent staining, the BMSCs co-incubated with scaffolds were stained with 4',6-diamidino-2-phenylindole (DAPI, blue) and rhodamine-phalloidin (red) to directly observe subtle cytoskeletal changes under a LEICA microscope.

**Animal experiments**. With the approval from the Animal Research Committee at Shanghai Sixth People's Hospital, 20 8-week-old male Sprague-Dawley rats (250 g) were acquired and randomly divided into three groups for animal experiments. Chloral hydrate sodium (250 mg·kg$^{-1}$) administered by intraperitoneal injection was used to anaesthetize rats. A 1.5–2.0 cm sagittal incision was cut on the scalp, then an electric trephine was used to make two full-thickness 5-mm diameter defects on both sides of the skull. The scaffolds (5 mm in diameter and 2 mm in thickness) were randomly implanted into the cranial defects. After surgery, silk sutures were used to carefully suture the soft tissues. All rats were sacrificed at week 12 and the calvarias were harvested.

**Micro-CT scanning and histomorphometry**. The rat skulls were scanned by a micro-CT scanner (Skyscan, Kontich, Belgium). Tomograms were reconstructed with the 3D creator software (Skyscan), and the CTAn image analysis software was used to calculate the BV/TV ratio and the BMD (g·cm$^{-3}$) according to the reconstructed images. Next, the samples were decalcified, embedded, and cut (5-μm sections). The sections were then stained with H&E or Masson trichrome, or immunohistochemical staining was performed (NPFFR1, CD31, VEGF, and OPN). A LEICA microscope was used to capture all images.

**Statistics and reproducibility**. All data are presented as means ± standard error of the mean (S.E.M.). Student's t-test or one-way ANOVA was used to determine the differences between groups with Bonferroni's correction in SPSS 18 (IBM, Armonk, NY). *$P < 0.05$ was considered statistically significant. All the data in this article is reproducible. All experiments were of at least three replicates with detailed information listed in each experiment.

**Reporting summary**. Further information on research design is available in the Nature Portfolio Reporting Summary linked to this article.

## Data availability
The main data for the findings of this research are available within the Article and its Supplementary Information. The raw data supportinig the figures of this study are provided as Supplementary Data file, including all original western blot images. The newly generated plasmids are listed in Supplementary Table S2, with links provided to their complete sequences. Additional data are available from the corresponding author upon request.

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

## Acknowledgements

This project was partially funded by the National Key Research and Development Program of China (2020YFA0908100), the National Science Fund for Distinguished Young Scholars (Grant No. 32125023), the Joint Funds of the National Natural Science Foundation of China (Key Program No. U1932204), the National Natural Science Foundation of China (Grant No. 81902237) and the China Postdoctoral Science Foundation (Grant No. 2021M703380).

## Author contributions

Y.C., C.Z., and Y.G. conceived the concept and designed the experiments. H.Y. performed cell culture, angiogenesis assays, osteogenesis assays, and animal experiments; Y.W. performed protein coating design, characterization, and scaffold preparation; J.G. participated in microCT analysis and the relevant data statistics; Y.G. contributed to data analysis of in vitro tests; all the authors contributed to the manuscript writing.

## Competing interests

The authors declare no competing interests.
