## [Peer Review File · Communications Biology]

Reviewers' comments:

Reviewer #1 (Remarks to the Author):

The paper dealt with an interesting subject which illustrated the role of the neuropeptide NPVF in enhancing both angiogenesis and osteogenesis, and developed an NPVF peptide-fused amyloid coating method to enhance angiogenic osteogenesis. The combination of miRNA sequencing and logical experiments confirmed the function and features of NPVF peptide. The article was well written, it clearly reported the main outcomes of the paper. Nevertheless, some minor changes should be properly addressed by the authors to improve the clarity and the understanding.

- 1.The NPVF peptide-fused amyloid coating was physically combined with scaffolds, which could lead to an unstable degradation and release behavior, authors should give it a description and release behavior study as it is a crucial factor for biomaterials.
- 2.The nanofiber structure and morphology of NPVF peptide should be characterized.
- 3.All error lines were too light and fine to see clearly.
- 4.Please check the character abbreviation when it appeared in the article firstly, eg line 219 IHC.

Reviewer #2 (Remarks to the Author):

The authors of the article #COMMSBIO-22-1603-T titled "Viable Application of a Novel Neuropeptide for Enhancing Angiogenesis and Osteogenesis in Bone Regeneration" describe their investigation of the neuropeptide NPFV and its mechanisms in brain-to-bone regulation system and its potential for bone regeneration. The study describes a very interesting interdisciplinary approach to study – on different levels – the cross-talk between neuropeptides and bone and the associated skeletal homeostasis. Further the authors connect their findings with a proof-of-concept experiment for a physiological relevance and in vivo-effect in an animal model treating possible bone defects (calvarial bone) with hydroxyapatite scaffolds that are coated with the respective neuropeptide NPFV linked to a CsgA amyloid nanofiber, and demonstrated a positive effect on regeneration through enhanced angiogenesis. However, a few open questions regarding hypothesis and conclusions remain; therefore, in my opinion, some amendments will be required in order to make this manuscript acceptable for publication.

Introduction:

1) The initial idea and hypothesis should be phrased a bit clearer, as the authors state NPVF as a "representative of NPFF" (p.3/l.64) and the idea as "tantalizing" (p.3/l.61). It is not obvious how this NP was selected among other candidates such as NPSF or NPAF (was some pre-screening involved?), or whether these could play a role too (maybe add to discussion section).

2) hydroxyapatite is officially introduced with the abbreviation "HA", later "HAP" is used as an abbreviation in the results section below. Please keep it consistent throughout the manuscript. I would suggest "HAp".

Results:

3) p.5/l.93 ff: The authors describe a positive effect by NPFV concentrations (0.1-1 nM) on HUVEC proliferation (Fig.2B). However, apparently no statistically significant difference was detected, therefore the data interpretation here needs to be re-phrased accordingly.

4) p.6/l.132: besides miR 181c, which others of the upregulated miRNAs could play a substantial role? Why did you decide for miR-181c particularly for your investigation?

5) Please add scale bars to Figure 4D.

6) Fig.5: Isopropyl β -D-1-thiogalactopyranoside should be added to the figure caption as full expression for the abbreviation "IPTG". Should a negative control be considered for Congo-Red staining and THT assay in Fig.5B/C?

7) Fig.S3: cells and scale bars are very difficult to see, especially in the fluorescence images.

8) Figure 6A, D, E, F, G: please describe the qualitative histological analysis a bit more detailed and more comprehensively in the text (position of the scaffold; distribution of bone formation, scaffold remodelling, CD31 expression etc.).

Discussion:

9) p.12/l.260 "outperforming the existing xenogenous cell laden tissue engineering strategy, which frequently employs stem cells and growth factors and which unavoidably leads to various immune rejection and infection problems." I do not think that this rather critical conclusion can be drawn from the presented in vivo experiment and without any reference to previous studies that the authors might be referring to with frequently applied xenogenous stem cells leading to unavoidable immune rejections and infections. At least the authors should give clear examples of what they refer to.

10) In addition to the really interesting bone regenerative approach: On the other hand, how can this knowledge be used for further understanding in neuroscience?

11) After CsgA-NPVF nanofiber-coated scaffold application: would the authors also expect a similar positive angiogenic and regenerative effect in load-bearing/load-sharing bones and joints – other than the calvarial defect model?

Materials & Methods:

12) Maybe for the rat model, the term "clinical" should be replaced by "pre-clinical with a potential for clinical application later" instead.

13) The time points for the sample collection and experimental conditions should be also briefly mentioned in the results section and the respective figure captions, e.g. tube formation observed after 6 h, transwell incubation 24 h, VEGF ELISA after 48 h of VEGF secretion in the supernatant,...

Reviewer #3 (Remarks to the Author):

Yu et al. reported in their manuscript about the effect of the neuropeptide NPVF to enhance both angiogenesis and osteogenesis and figured out potential signaling pathways involved in this regulation. Moreover, they fused the NPVF with an amyloid-protein derived from E. coli biofilms, by genetical engineering, to improve its stability and bioavailability. This engineered proteins were used to coat hydroxyapatite-scaffolds exploiting a self-assembling process. The coated scaffolds were implanted in a rat calvaria defect model and analysed in comparison to a non-coated scaffold group and to scaffolds loaded only with the amyloid-protein regarding blood vessel and bone tissue ingrowth.

The study in general is well conducted albeit a few questions are not covered. The data provide novel insights. The manuscript is well and clearly written. The necessary background is given and the experimental work is understandable described, the work is clearly structured; a view points need improvement:

Questions:

1. Why did the authors not measure ALP activity and Ca deposition quantitatively? These are simple assays and would provide quantitative data which are stronger than the qualitative stainings.

2. Does the low number of rats per group allow to draw valid conclusions or is it rather a pilot study? Please clarify.

3. As the amyloid-protein is of bacterial origin: does it provoke reactions from the immune

system/inflammation? The authors should comment on that.

4. It would have been helpful to analyse the coated scaffolds first in vitro to explore cytocompatibility of the coating and if the efficacy of the hybrid-protein is similar to the pure NPVF. The authors should discuss that.

5. Another limitation of the work is in my eyes that there are no experimental data about the release of NPVF from the coating. Is the factor released or is it also effective in bound form? At least, this issue should be discussed.

Comments regarding improvement:

1. 20 rats were used for three groups – how many animals for which group? This should be clarified.

2. There is no exact information about the 3d HA scaffolds (in the methods: source/company) and also the description on lines 276-278 is very general. It is necessary to add specific details about the scaffolds (material characteristics, fabrication process ...).

3. Figure 2: the font size of the oft he labeling oft he axes in the diagrams are too small.

4. Figure 3, 4, 6: the images and diagrams are too small and also the font sizes

5. Figure 6/caption: 5 samples in B and C – does this mean 5 animals? Please clarify.

6. Scale bars are hardly readable and sometimes missing.

7. Lines 324 and 348: „conditional medium“ – it is not clear that is meant. Please clarify.

8. Line 407: the term „cocultured“ should be replaced as it implies the usage oft wo cell types. The scaffold was seeded or colonized with the cells. Or were the cells cultured next to the scaffolds? Please clarify.

9. In the discussion, there is some avavoidable repetition from the results part.

Nov 20, 2022

Re: Viable Application of a Novel Neuropeptide for Enhancing Angiogenesis and Osteogenesis in Bone Regeneration

SUMMARY:

We would first like to thank the editor and the three reviewers for their helpful and constructive input about our manuscript. Before getting into the full point-by-point response, we would like to summarize the key aspects of our revision experiments and our redrafting of the manuscript (with changes marked in **blue color** in the main text). Following the reviewer's helpful suggestion, we have conducted *in vitro* release experiments and added a discussion in the revised manuscript. Specific experimental details and discussions are given in the point-by-point response below.

1. We had made clarifications regarding the methods and results section in the revised manuscript including the coating process, Cell Counting Kit-8 (CCK-8) assay, Hematoxylin & eosin (H&E) staining and Masson staining.
2. We have justified the type and number of animals used in our experiments design, we provide our explanations as follows: (1) our experiment designs strictly follow the 3Rs (Replacement, Reduction and Refinement) of the Principles of Humane Experimental Technique, which aim to provide a framework to ensure that animal research was undertaken as humanely as possible. Specifically, the definition of Reduction is to minimize the number of animals for per study (e.g, making multiple measurements, and at multiple time points, from the same animal). (2) the number of animals used in our experimental designs allowed us to perform appropriate statistical analysis and draw a conclusion based on the data. (3) the animal model was a pilot study in the present article. We plan to translate the CsgA-NPVF nanofiber-coated scaffold into clinical practice in future studies including evaluation of various properties of the CsgA-NPVF nanofiber-coated scaffold and adopt other animal models suitable for clinical application, i.e., load-bearing/load-sharing bone and joint models, and by then we'll include large animal tests.
3. We have tested the stability of the protein coating by *in vitro* release experiments and added a discussion in the revised manuscript. Specifically, we immersed the CsgA-NPVF-coated HAp scaffolds in a simulated body fluid (SBF) to assess their *in vitro* degradation and release behaviors.
4. We have performed TEM imaging to reveal and compare the morphologies of NPVF and CsgA-NPVF proteins, revealing that only CsgA-NPVF proteins exhibited nanofiber morphologies.
5. We have supplemented the corresponding control experiments for Congo-Red staining and THT assay. The imidazole buffer used to elute the target proteins during protein purification was used as a control for Congo red staining and ThT fluorescence assay.

6. We had measured the ALP activity and Ca deposition quantitatively. The quantification of Alizarin red staining was performed according to the protocol of Hexadecylpyridinium chloride monohydrate (Solarbio, Beijing, China), the OD value was detected at 562 nm by a multifunctional microplate reader after the dissolution of mineralization nodes.
7. We have optimized the figures to be more aesthetically pleasing and accessible, such as bolding the scalebar, correction of typos, and adjustment of images.

Point-by-point response:

Reviewer #1 (Remarks to the Author):

The paper dealt with an interesting subject which illustrated the role of the neuropeptide NPVF in enhancing both angiogenesis and osteogenesis, and developed an NPVF peptide-fused amyloid coating method to enhance angiogenic osteogenesis. The combination of miRNA sequencing and logical experiments confirmed the function and features of NPVF peptide. The article was well written, it clearly reported the main outcomes of the paper. Nevertheless, some minor changes should be properly addressed by the authors to improve the clarity and the understanding.

We thank the reviewer for the positive and constructive comments.

1. The NPVF peptide-fused amyloid coating was physically combined with scaffolds, which could lead to an unstable degradation and release behavior, authors should give it a description and release behavior study as it is a crucial factor for biomaterials.

Reply: The reviewer raised a very important point that we haven't paid enough attention before. Following the reviewer's suggestion, we have tested the stability of the protein coating by *in vitro* release experiments and added a discussion in the revised manuscript. Specifically, we immersed the CsgA-NPVF-coated HAp scaffolds in a simulated body fluid (SBF) to assess their *in vitro* degradation and release behaviors. The release of His-tag labelled CsgA-NPVF proteins in the SBF solution at different time points was monitored using a commercially available enzyme-linked immunosorbent assay (ELISA) kit. During the 60-day test, the amount of proteins released into the solution in the CsgA-NPVF groups was almost the same as that in the control group (uncoated HA scaffolds), indicating that the CsgA-NPVF coating had good stability (**Fig. R1**).

Fig. R1 Protein release curve of CsgA-NPVF-coated scaffolds and uncoated scaffolds in SBF solution.

Revisions in the main text:

To further explore the stability of the recombinant protein coating, we performed *in vitro* protein release experiments in simulated body fluid (SBF). Both uncoated scaffolds and CsgA-NPVF-coated scaffolds were immersed in SBF solution (pH=7.4) and incubated at 37 °C for two months. His-tag labelled CsgA-NPVF proteins released into the SBF solution was quantitatively characterized using a commercially available enzyme-linked immunosorbent assay (ELISA) kit. During the 60-day test, the amount of supernatant protein in the CsgA-NPVF groups was almost the same as that in the control group (Fig. S5), indicating that the CsgA-NPVF coating had good stability.

Methods

***In vitro* protein release experiment**

After the coating process, both uncoated HAp scaffolds and CsgA-NPVF-coated scaffolds were immersed in 10 mL SBF solution (pH=7.4) and incubated at 37 °C for two months. The amount of His-tag labelled CsgA-NPVF proteins released into the solution were detected using a commercially available His tag ELISA detection kit (Genscript, Nanjing, China). ELISA experiments were performed according to the instruction at 1, 3, 7, 14, 30 and 60 days after immersion.

2. The nanofiber structure and morphology of NPVF peptide should be characterized.

Reply: Thanks to the reviewer for pointing this out. NPVF is an eight amino acid polypeptide that doesn't form a fibrous structure by itself. Note that NPVF-containing nanofibers can only be formed when expressed as a fusion protein (CsgA-NPVF). The morphologies of NPVF and CsgA-NPVF proteins assessed by TEM imaging were provided below as a reference (Fig. R2).

Fig. R2 TEM images of NPVF polypeptide and CsgA-NPVF nanofibers.

Revisions in the main text:

After induction of bacterial expression and protein purification, we demonstrated by Transmission Electron Microscopy (TEM) that NPVF-containing nanofibers could only be formed when expressed in fusion with CsgA (**Fig. S4**).

Methods

Transmission Electron Microscopy (TEM) imaging

For TEM imaging, a 10 μ L droplet of protein was directly deposited on TEM grids (Zhongjingkeyi Technology, EM Sciences) for 5 min. The excessive solution was wicked away with pieces of filter paper and the samples were rinsed twice with ddH₂O by placing 10 μ L ddH₂O on the TEM grids and quickly wicking off with filter paper. The samples were then negatively stained with 10 μ L 2 wt% uranyl acetate for 1 min and dried for 15 min under an infrared lamp. TEM images were obtained on a Tecnai G2 Spirit T12 transmission electron microscope operated at 120 kV accelerating voltage.

3. All error lines were too light and fine to see clearly.

Reply: Thank you very much for the helpful comments. We had made necessary changes in the error lines in the revised manuscript.

4. Please check the character abbreviation when it appeared in the article firstly, eg line 219 IHC.

Reply: Thank you very much for the helpful comments. Following your suggestion, we had doubled checked the character abbreviation, and made sure that all the full names of the abbreviations were added (e.g. immunohistochemistry (IHC)) in the revised manuscript when they firstly appeared in the article.

Reviewer #2 (Remarks to the Author):

The authors of the article #COMMSBIO-22-1603-T titled “Viable Application of a Novel Neuropeptide for Enhancing Angiogenesis and Osteogenesis in Bone Regeneration” describe their investigation of the neuropeptide NPVF and its mechanisms in brain-to-bone regulation system and its potential for bone regeneration. The study describes a very interesting interdisciplinary approach to study – on different levels – the cross-talk between neuropeptides and bone and the associated skeletal homeostasis. Further the authors connect their findings with a proof-of-concept experiment for a physiological relevance and in vivo-effect in an animal model treating possible bone defects (calvarial bone) with hydroxyapatite scaffolds that are coated with the respective neuropeptide NPVF linked to a CsgA amyloid nanofiber, and demonstrated a positive effect on regeneration through enhanced angiogenesis. However, a few open questions regarding hypothesis and conclusions remain; therefore, in my opinion, some amendments will be required in order to make this manuscript acceptable for publication.

We thank the reviewer for the positive and constructive comments.

Introduction:

1. The initial idea and hypothesis should be phrased a bit clearer, as the authors state NPVF as a “representative of NPFF” (p.3/l.64) and the idea as “tantalizing” (p.3/l.61). It is not obvious how this NP was selected among other candidates such as NPSF or NPAF (was some pre-screening involved?), or whether these could play a role too (maybe add to discussion section).

Reply: The reviewer raised very good points. Neuropeptide FF (NPFF, FLFQPQRF-NH₂) belongs to the RF-amide neuropeptide family, and is first identified from bovine brain extract by isolation of the endogenous peptides which had cross immunoreactivity with cardio-excitatory peptide FMRF-NH₂ in 1985¹. Indeed, NPAF (NPSF, consisting 18 amino acids, had been used in the literature to describe both the short version of NPAF (8 amino acids) and an extended version of RFRP-1(37 amino acid) is an also important family member of NPFF system. We initially chose the shorter polypeptide (NPVF) in our study rather than the longer polypeptide because we rationalize that the fusion of the shorter polypeptide onto CsgA would not disrupt the self-assembly of CsgA protein and thus ensure their fiber coating formation abilities. In addition, our series of previous works mainly focused on the osteo-inductivity of NPVF and the in-depth molecular mechanism (manuscript to be submitted), the selected NPVF has already been validated with specific functions and perfectly met the need of current studies. It should be noticed that neither NPSF nor NPAF should be ignored or excluded from the future study of NPFF family, where they might also play critical role.

Revisions in the main text:

Note that NPAF (three versions, including the NPSF peptide containing 18 amino acid

residues, the 8 amino acid short version of NPAF and a 37 amino acid extended version of RFRP-1, had been described in the literature) is also an important family member of NPFF system (21, 61). We initially chose the shorter polypeptide (NPVF) in our study rather than the longer polypeptide (NPSF) because we rationalize that the fusion of the shorter polypeptide onto CsgA would not disrupt the self-assembly of CsgA protein and thus ensure their fiber coating formation abilities. In addition, our series of previous works mainly focused on the osteo-inductivity of NPVF and the in-depth molecular mechanism (manuscript to be submitted), the selected NPVF had already been validated with specific functions and perfectly met the need of current studies. However, it should be noticed that neither NPSF nor NPAF should be ignored or excluded from the future study of NPFF family, where they might also play critical role. It should be noted that the NPVF itself is an important member in the family of neuropeptide. This work aimed at introducing the possibility of utilizing NPVF in skeletal system which widens the application of neuropeptides. Combined with our previous findings that NPFF activation of NPFF2 receptor stimulates neurite outgrowth in Neuro 2A cells via activating the ERK signaling pathway (18), we will further work on how this neuropeptide regulates bone metabolism and the feedback of skeletal system to the brain in the future.

2. Hydroxyapatite is officially introduced with the abbreviation “HA”, later “HAP” is used as an abbreviation in the results section below. Please keep it consistent throughout the manuscript. I would suggest “HAp”.

Reply: Thank you very much for your careful review. We had changed the “HA” for “HAp” in the revised manuscript.

3. p.5/l.93 ff: The authors describe a positive effect by NPVF concentrations (0.1-1 nM) on HUVEC proliferation (Fig.2B). However, apparently no statistically significant difference was detected, therefore the data interpretation here needs to be re-phrased accordingly.

Reply: Thank you very much for your careful review. We had rephrased the effect of NPVF concentrations (0.1-1 nM) on HUVECs proliferation in the revised manuscript.

Revisions in the main text:

A Cell Counting Kit-8 (CCK-8) assay showed that low dose of NPVF (0.01, 0.1 and 1 nM) had no toxic effect on the proliferation of HUVECs, while high dose of NPVF (10 nM) had a slight inhibition on the proliferation of HUVECs (**Fig. 2B**).

4. p.6/l.132: besides miR 181c, which others of the upregulated miRNAs could play a substantial role? Why did you decide for miR-181c particularly for your investigation?

Reply: Thank you very much for the helpful comments. The miRNA-seq results showed that six miRNAs were upregulated and thirteen miRNAs were downregulated in the NPVF-treated HUVECs compared with the untreated HUVECs (**Fig. 3A**). Indeed, all those upregulated miRNAs might play substantial role for HUVECs, we only choose

miR-181c in our study as previous studies have shown that target genes of miR-181c-3p played a substantial role in endothelial cell function^{4,5}; and we would like to focus on one so that we can possibly gain insights into the mechanism. We will explore the mechanism of other miRNAs in the future to better understand the overall regulation effect of NPFF system on bone. We have accordingly made some necessary explanations in the main text.

Revisions in the main text:

The miRNA-seq results showed that six miRNAs were upregulated and thirteen miRNAs were downregulated in the NPVF-treated HUVECs compared with untreated HUVECs (Fig. 3A). Among all the miRNAs with upregulated expression, we were particularly interested in miR-181c, the target gene of which plays a substantial role in endothelial cell function (32, 33).

5. Please add scale bars to Figure 4D.

Reply: Thank you very much for your careful review. We have added scale bars to Figure 4D in the revised manuscript.

Revisions in the main text:

Fig. 4. The WNT/β-catenin pathway mediates the NPVF-induced osteogenic differentiation of BMSCs.

6. Fig.5: Isopropyl β-D-1-thiogalactopyranoside should be added to the figure caption as full expression for the abbreviation “IPTG”. Should a negative control be considered for Congo-Red staining and THT assay in Fig.5B/C?

Reply: Thank you very much for your helpful suggestions. We have added the full name of IPTG in the figure caption (**Figure 5**). In addition, we have supplemented the corresponding control experiments for Congo-Red staining and ThT assay as suggested by the reviewers. The imidazole buffer used to elute the target proteins during protein purification was used as a control for Congo red staining and ThT fluorescence assay.

Revisions in the main text:

Fig. 5 Preparation and characterization of the CsgA-NPVF nanofiber coating. (A) Schematic diagram depicting the induction of recombinant protein expression by *Isopropyl β -D-1-thiogalactopyranoside* (IPTG) and the preparation of nanofiber coatings on a porous 3-D hydroxyapatite (HAp) scaffold. (B) Congo red staining of the purified CsgA and CsgA-NPVF nanofibers. (C) Assembly kinetics of CsgA and CsgA-NPVF monomer solutions monitored by ThT fluorescence assay. *The imidazole buffer used to elute the target proteins during protein purification was used as a control for Congo red staining and ThT fluorescence assay.* (D) AFM images showing the morphology of the self-assembled protein nanofibers. (E) Comparison of the adsorption behavior of the CsgA and CsgA-NPVF nanofibers on HAp substrates measured by QCM. (F) SEM images showing the surface morphology of the 3D HAp scaffold before and after the coating process.

7. Fig.S3: cells and scale bars are very difficult to see, especially in the fluorescence images.

Reply: Thank you very much for suggestion. We had adjusted the resolution of the pictures in the revised manuscript, such as **Fig. 2**.

8. Figure 6A, D, E, F, G: please describe the qualitative histological analysis a bit more detailed and more comprehensively in the text (position of the scaffold; distribution of bone formation, scaffold remodelling, CD31 expression etc.).

Reply: Thank you very much for your helpful comments. We had described the qualitative histological analysis of Figure 6 in details in the revised manuscript.

Revisions in the main text:

Furthermore, at 12 weeks after transplantation, micro-CT reconstruction data indicated that more fresh bone formation formed into the direct 3D space of the CsgA-NPVF protein-coated scaffolds in the defects than in those treated with the control groups (bare HAp scaffolds and CsgA protein-coated scaffolds) (**Fig. 6A**). Quantitative analysis revealed that both the bone mineral density (BMD, 1.538 g/cm³ VS 1.276 g/cm³) and the bone volume/tissue volume (BV/TV, 69.069% VS 50.142%) ratio in the CsgA-NPVF group were higher than those in the control groups (**Fig. 6B-C**). Hematoxylin & eosin (H&E) staining and Masson staining all demonstrated enhanced osteoblastic marker expression and bone matrix formation occurred in the 3D space of the CsgA-NPVF protein-coated scaffolds in the defects than in those treated with the control groups (bare HAp scaffolds and CsgA protein-coated scaffolds (**Fig. 6D-E**). Moreover, IHC analysis revealed much higher expression levels of CD31 and COL I in the bone matrix formation for the CsgA-NPVF group (**Fig. 6F-G**).

Discussion:

9. p.12/l.260 “outperforming the existing xenogenous cell laden tissue engineering strategy, which frequently employs stem cells and growth factors and which unavoidably leads to various immune rejection and infection problems.” I do not think that this rather critical conclusion can be drawn from the presented in vivo experiment and without any reference to previous studies that the authors might be referring to with frequently applied xenogenous stem cells leading to unavoidable immune rejections and infections. At least the authors should give clear examples of what they refer to.

Reply: We thank the reviewer’s constructive suggestion. We have made corresponding changes to draw a more solid conclusion.

The main advantages of peptides are their high target affinity, safety, targeting a wide range of molecules, and low toxicity, immunogenicity⁶⁻⁸. They may perform better in some aspects than commonly used xenogenous cell-laden tissue engineering strategy, which frequently meets problems including immune rejection and donor-donor variability⁹⁻¹¹. Notably, NPVF represents a new type of endogenous peptide that exhibits promising therapeutic efficacy. The functional coatings based on the NPVF

peptide-containing fusion proteins might avoid problems including immune rejection and donor-donor variability where existing xenogenous cell-laden tissue engineering strategy frequently meets ^{12, 13}.

First, it is acknowledged that the host's immune system starts to attack the xenogenous cells once transplantation is done, unless immunosuppress were utilized. This is also the reason why xenogenous cell transplantation are performed on athymic animals. Second, for example, MSCs as one of the most popular transplanted cells, are known to exhibit donor-donor variability, to eliminate the difference between donors, and all experiments should be performed using a single donor¹¹.

Revisions in the main text:

Notably, NPVF represents a new type of endogenous peptide that exhibits promising therapeutic efficacy. The main advantages of peptides include their safety, the high target affinity, targeting a wide range of molecules, low toxicity and low immunogenicity⁶⁻⁸. The functional coatings based on the NPVF peptide-containing fusion proteins might avoid problems including immune rejection and donor-donor variability where existing xenogenous cell-laden tissue engineering strategy frequently meets ^{12, 13}.

10. In addition to the really interesting bone regenerative approach: On the other hand, how can this knowledge be used for further understanding in neuroscience?

Reply: Thank you very much for your insightful comments. We have added a discussion paragraph in the main text to illustrate the potential implications of our work for neuroscience.

Revisions in the main text:

It should be noted that the NPVF itself is an important member in the family of neuropeptide. This work aimed at introducing the possibility of utilizing NPVF in skeletal system which widens the application of neuropeptides. Combined with our previous findings that NPFF activation of NPFF2 receptor stimulates neurite outgrowth in Neuro 2A cells via activating the ERK signaling pathway ¹⁴, we will further work on how this neuropeptide regulates bone metabolism and the feedback of skeletal system to the brain in the future.

11. After CsgA-NPVF nanofiber-coated scaffold application: would the authors also expect a similar positive angiogenic and regenerative effect in load-bearing/load-sharing bones and joints – other than the calvarial defect model?

Reply: Thank you for the insightful comments. We believe that the CsgA-NPVF nanofiber-coated scaffold might exhibit similar angiogenic and regenerative effect in load-bearing bones, not just in calvarial bone. However, we should strengthen that it indeed requires additional experiments and design improvement in term of mechanical strength (including the design of coating on plates or intramedullary nails) to support the body weight. Those experiments would be particularly important for future translation of this technology into clinical application.

Materials & Methods:

12. Maybe for the rat model, the term “clinical” should be replaced by “pre-clinical with a potential for clinical application later” instead.

Reply: We had replaced the term “clinical” with “pre-clinical” in the revised manuscript.

13. The time points for the sample collection and experimental conditions should be also briefly mentioned in the results section and the respective figure captions, e.g. tube formation observed after 6 h, transwell incubation 24 h, VEGF ELISA after 48 h of VEGF secretion in the supernatant.

Reply: We thank the reviewer for the constructive comments. We had added the time points for the sample collection and experimental conditions in the revised manuscript.

Reviewer #3 (Remarks to the Author):

Yu et al. reported in their manuscript about the effect of the neuropeptide NPVF to enhance both angiogenesis and osteogenesis and figured out potential signaling pathways involved in this regulation. Moreover, they fused the NPVF with an amyloid-protein derived from E. coli biofilms, by genetical engineering, to improve its stability and bioavailability. These engineered proteins were used to coat hydroxyapatite-scaffolds exploiting a self-assembling process. The coated scaffolds were implanted in a rat calvaria defect model and analysed in comparison to a non-coated scaffold group and to scaffolds loaded only with the amyloid-protein regarding blood vessel and bone tissue ingrowth.

The study in general is well conducted albeit a few questions are not covered. The data provide novel insights. The manuscript is well and clearly written. The necessary background is given and the experimental work is understandable described, the work is clearly structured; a view of points need improvement:

Questions:

1. Why did the authors not measure ALP activity and Ca deposition quantitatively? These are simple assays and would provide quantitative data which are stronger than the qualitative stainings.

Reply: Thank you very much for your helpful comments. We had measured the ALP activity and Ca deposition quantitatively, but we did not include the data in the main figures in our initial submission. In general, the quantification of Alizarin red staining was performed according to the protocol of Hexadecylpyridinium chloride monohydrate (Solarbio, Beijing, China), the OD value was detected at 562 nm by a multifunctional microplate reader after the dissolution of mineralization nodes. The results were shown in **Fig. R3A**. The quantification of ALP staining was performed by an Alkaline phosphatase assay kit (Jiancheng, Nanjing, China) in the manuscript. The results were shown in **Fig. R3B** (In the manuscript, the staining images and

corresponding quantitative results were shown in Figure 4D and Figure 3S, respectively).

Revisions in the main text:

Materials and methods

The quantification of Alizarin red staining was performed according to the protocol of Hexadecylpyridinium chloride monohydrate (Solarbio, Beijing, China), the OD value was detected at 562 nm by a multifunctional microplate reader after the dissolution of mineralization nodes. The quantification of ALP staining was performed by an Alkaline phosphatase assay kit (Jiancheng, Nanjing, China) in the manuscript.

Results

Alizarin Red S (ARS, 21 days) and alkaline phosphatase (ALP, 7 days) assays were carried out to explore the role of the Wnt/ β -catenin pathway in NPVF-induced osteogenic differentiation of BMSCs. Both the staining and corresponding quantitative results indicated that NPVF remarkably promoted extracellular mineralization and ALP activity in BMSCs, while these effects were abolished in the presence of either RF9 or JW74 (Fig. 4D, Fig. S3). In conclusion, these results revealed that NPVF promotes osteogenic differentiation of BMSCs via the Wnt/ β -catenin signaling pathway.

A

B

Fig. R3 The quantification of ARS and ALP staining. * $p < 0.05$.

2. Does the low number of rats per group allow to draw valid conclusions or is it rather a pilot study? Please clarify.

Reply: Thank the review's constructive suggestion. Indeed, the animal model here only serves as a pilot study. Our next research plan is to translate CsgA-NPVF nanofiber-coated scaffold into clinical practice, which must evaluate various properties of the CsgA-NPVF nanofiber-coated scaffold and adopt other animal models suitable for clinical application, i.e., load-bearing/load-sharing bones and joints models.

3. As the amyloid-protein is of bacterial origin: does it provoke reactions from the immune system/inflammation? The authors should comment on that.

Reply: We fully understand the reviewer's concerns. Although derived from bacteria, CsgA protein has been used in various medical applications, such as surface modification of cell culture scaffolds and bone grafts¹⁵, secretion of CsgA-fused cytokine for the treatment of inflammatory bowel disease (IBD)¹⁶. Previous studies have shown that CsgA protein is not only non-pathogenic, but can even promote cell adhesion and proliferation. We have added a description towards the safety of CsgA amyloid protein in the revised manuscript. Besides, *in vitro* cell culture experiments have been conducted to verify the biocompatibility of CsgA-NPVF-coated scaffolds (**Figure S4**). However, if it involves clinical applications, its long-term safety and stability need to be further evaluated.

Revisions in the main text:

Owing to its good biocompatibility and surface adhesion capacity, CsgA have been used in various medical applications, such as surface modification of cell culture scaffolds and bone implants¹⁵, secretion of CsgA-fused cytokine for the treatment of inflammatory bowel disease (IBD)¹⁶.

4. It would have been helpful to analyse the coated scaffolds first *in vitro* to explore cytocompatibility of the coating and if the efficacy of the hybrid-protein is similar to the pure NPVF. The authors should discuss that.

Reply: Thank you very much for your helpful comments. The cytocompatibility of coated scaffolds was assessed using BMSCs and the results indicated that compared with cells grown on the empty HA scaffolds and CsgA-coated scaffolds, cells grown on the CsgA-NPVF-coated scaffolds had the best adhesion capacity and the highest viability (**Fig. S6**), which demonstrated that coated scaffolds possessed good adhesion capacity and had no cytocompatibility. Moreover, the CCK-8 assay indicated that low dose of NPVF (0.01, 0.1 and 1 nM) had no toxic effect on the proliferation of HUVECs (**Fig. 2B**). Taken together, these results thus indicate the CsgA-NPVF-coated scaffolds had good adhesion capacity and no cytocompatibility.

As reported earlier, the bioavailability of free short peptides inside a body is usually hampered by their structural vulnerability in the serum environment and rapid clearance via renal filtration. The pure NPVF peptides tend to freely diffuse out of the scaffolds and can thus suffer from such issues as well. In addition, the potential leakage of the peptides will likely result in random bond formation rather than specific bone regeneration in the scaffolds. Taken together, we believe that the efficacy of the pure NPVF might be similar to that of the fusion protein in terms of bone formation, but they would not achieve site-specific bone formation as the protein coatings do.

Revisions in the main text:

Owing to its good biocompatibility and surface adhesion capacity, CsgA has been used in various medical applications, such as surface modification of cell culture scaffolds and bone implants (34), secretion of CsgA-fused cytokine for the treatment of inflammatory bowel disease (IBD) (35). Based on previous studies, we reasonably speculated that fusion with CsgA could improve the stability of NPVF peptides while preserving their biological activity. Before animal experiments, we tested the cytocompatibility of different scaffolds using BMSCs and found that compared with cells grown on the empty HAp scaffolds and CsgA-coated scaffolds, cells grown on the CsgA-NPVF-coated scaffolds had the best adhesion capacity and the highest viability (**Fig. S6**). We next evaluated the efficacy of the NPVF-containing protein-coated HAp scaffolds in assisting bone regeneration *in vivo* using a standard rat calvarial defect model.

5. Another limitation of the work is in my eyes that there are no experimental data about the release of NPVF from the coating. Is the factor released or is it also effective in bound form? At least, this issue should be discussed.

Reply: In our revised manuscript, we have tested the stability of the protein coating by *in vitro* release experiments and added a discussion in the revised manuscript. Due to the lack of advanced characterization techniques, it's difficult for us to trace where these recombinant proteins go *in vivo*. Therefore, we immersed the CsgA-NPVF-coated HAp scaffolds in simulated body fluid (SBF) for two months to study their *in vitro* degradation and release behaviors. His-tag labelled CsgA-NPVF proteins in the SBF solution at different time points was detected using a commercially available enzyme-linked immunosorbent assay (ELISA) kit. During the 60-day test, the amount of supernatant protein in the CsgA-NPVF groups was almost the same as that in the

control group (uncoated HA scaffolds), indicating that the CsgA-NPVF coating had good stability (**Fig. R1**).

Fig. R1 Protein release curve of CsgA-NPVF-coated scaffold and uncoated scaffold in SBF solution.

Revisions in the main text:

To further explore the stability of the recombinant protein coating, we performed *in vitro* protein release experiments in simulated body fluid (SBF). Both uncoated scaffolds and CsgA-NPVF-coated scaffolds were immersed in SBF solution (pH=7.4) and incubated at 37 °C for two months. His-tag labelled CsgA-NPVF proteins released into the SBF solution was quantitatively characterized using a commercially available enzyme-linked immunosorbent assay (ELISA) kit. During the 60-day test, the amount of supernatant protein in the CsgA-NPVF groups was almost the same as that in the control group (**Fig. S5**), indicating that the CsgA-NPVF coating had good stability.

Comments regarding improvement:

6. 20 rats were used for three groups – how many animals for which group? This should be clarified.

Reply: Thanks for your constructive suggestion. We have clarified corresponding descriptions: the blank scaffold group (n=6), CsgA scaffold group (n=7) and the CsgA-NPVF scaffold group (n=7).

The choosing of specific number of animals for each group was based on the following reasons. Firstly, our tests strictly follow the 3Rs (Replacement, Reduction and Refinement) of the Principles of Humane Experimental Technique, which aim to provide a framework to ensure that animal research was undertaken as humanely as possible. Specifically, the definition of Reduction is to minimize the number of animals for per study (e.g. making multiple measurements, and at multiple time points, from the

same animal), whilst maintaining robust experimental design to enable appropriate statistical analysis of the data.

Secondly, the animal model was a pilot study in the present article, because our next research plan is to translate CsgA-NPVF nanofiber-coated scaffold into clinical practice, which must evaluate various properties of the CsgA-NPVF nanofiber-coated scaffold and adopt animal models suitable for clinical application, i.e., load-bearing/load-sharing bones and joints models, and by then we'll perform a large animal study. Anyway, we appreciate your constructive suggestion very much.

7. There is no exact information about the 3d HA scaffolds (in the methods: source/company) and also the description on lines 276-278 is very general. It is necessary to add specific details about the scaffolds (material characteristics, fabrication process ...).

Reply: Thank you very much for your suggestion. We have supplemented the exact information of the 3D HAp scaffolds we use.

Revisions in the main text:

The 3D HAp scaffolds were purchased from INNOTERE company (111CC4), the scaffold consists of synthetic calcium phosphate (mainly α -tricalcium phosphate) and nanocrystalline calcium deficient hydroxyapatite. The scaffold offers interconnected porosity (3D version), high bioactivity, ease of handling, and high mechanical stability, making them ideal substrates for cell cultures in the field of bone regeneration.

8. Figure 2: the font size of the labeling of the axes in the diagrams are too small.

Reply: We had adjusted the font size of the labeling of the axes in the diagrams in Figure 2 in the revised manuscript.

9. Figure 3, 4, 6: the images and diagrams are too small and also the font sizes.

Reply: We had adjusted the images, diagrams, and the font sizes in the revised manuscript.

10. Figure 6/caption: 5 samples in B and C – does this mean 5 animals? Please clarify.

Reply: Thank you very much for your careful review. Indeed, 5 samples in B and C mean 5 animals. We replaced the expression of 5 samples with 5 animals.

11. Scale bars are hardly readable and sometimes missing.

Reply: We had adjusted the scale bars and added the missing scale bars in the revised manuscript.

12. Lines 324 and 348: “conditional medium”– it is not clear that is meant. Please clarify.

Reply: The conditional medium refers to a basal culture medium added with special drugs or reagents for specific cell or tissue culture, for example, ECM medium added

with NPVF, ECM medium supplemented with NPVF and RF9, ECM medium supplemented with miR-NC mimic, ECM medium supplemented with miR-181c-3p mimic, ECM medium supplemented with NC siRNA, ECM medium supplemented with AGO1 siRNA etc. We had used the exact descriptions in the revised manuscript.

13. Line 407: the term, “co-cultured” should be replaced as it implies the usage of two cell types. The scaffold was seeded or colonized with the cells. Or were the cells cultured next to the scaffolds? Please clarify.

Reply: Thank you very much for your helpful comments. We had replaced the “co-cultured” with “seeded” in the revised manuscript. Thank you again.

14. In the discussion, there is some avoidable repetition from the results part.

Reply: Thank you very much for your helpful comments. We had deleted the repetition from the results part in the discussion section in the revised manuscript.

Reference:

1. Yang HY, Fratta W, Majane EA and Costa E. Isolation, sequencing, synthesis, and pharmacological characterization of two brain neuropeptides that modulate the action of morphine. *Proc Natl Acad Sci U S A*. 1985;82:7757-61.
2. Mollereau C, Mazarguil H, Marcus D, Quelven I, Kotani M, Lannoy V, Dumont Y, Quirion R, Detheux M, Parmentier M and Zajac JM. Pharmacological characterization of human NPPF(1) and NPPF(2) receptors expressed in CHO cells by using NPY Y(1) receptor antagonists. *European journal of pharmacology*. 2002;451:245-56.
3. Bonini JA, Jones KA, Adham N, Forray C, Artymyshyn R, Durkin MM, Smith KE, Tamm JA, Boteju LW, Lakhani PP, Raddatz R, Yao WJ, Ogozalek KL, Boyle N, Kouranova EV, Quan Y, Vaysse PJ, Wetzel JM, Branchek TA, Gerald C and Borowsky B. Identification and characterization of two G protein-coupled receptors for neuropeptide FF. *J Biol Chem*. 2000;275:39324-31.
4. Zitman-Gal T, Green J, Pasmanik-Chor M, Golan E, Bernheim J and Benchetrit S. Vitamin D manipulates miR-181c, miR-20b and miR-15a in human umbilical vein endothelial cells exposed to a diabetic-like environment. *Cardiovascular diabetology*. 2014;13:8.
5. Sun X, Sit A and Feinberg MW. Role of miR-181 family in regulating vascular inflammation and immunity. *Trends in cardiovascular medicine*. 2014;24:105-12.
6. Fosgerau K and Hoffmann T. Peptide therapeutics: current status and future directions. *Drug Discov Today*. 2015;20:122-8.
7. Erak M, Bellmann-Sickert K, Els-Heindl S and Beck-Sickinger AG. Peptide chemistry toolbox - Transforming natural peptides into peptide therapeutics. *Bioorg Med Chem*. 2018;26:2759-2765.
8. Negahdaripour M, Owji H, Eslami M, Zamani M, Vakili B, Sabetian S, Nezafat N and Ghasemi Y. Selected application of peptide molecules as pharmaceutical agents and in cosmeceuticals. *Expert Opin Biol Ther*. 2019;19:1275-1287.
9. McDermott A, Herberg S, Mason D, Collins J, Pearson H, Dawahare J, Tang R, Patwa

- A, Grinstaff M, Kelly D, Alsberg E and Boerckel J. Recapitulating bone development through engineered mesenchymal condensations and mechanical cues for tissue regeneration. *Science translational medicine*. 2019;11.
10. Uyama H, Tu H, Sugita S, Yamasaki S, Kurimoto Y, Matsuyama T, Shiina T, Watanabe T, Takahashi M and Mandai M. Competency of iPSC-derived retinas in MHC-mismatched transplantation in non-human primates. *Stem cell reports*. 2022.
11. Solorio LD, Dhami CD, Dang PN, Vieregge EL and Alsberg E. Spatiotemporal regulation of chondrogenic differentiation with controlled delivery of transforming growth factor- β 1 from gelatin microspheres in mesenchymal stem cell aggregates. *Stem Cells Transl Med*. 2012;1:632-9.
12. McDermott AM, Herberg S, Mason DE, Collins JM, Pearson HB, Dawahare JH, Tang R, Patwa AN, Grinstaff MW, Kelly DJ, Alsberg E and Boerckel JD. Recapitulating bone development through engineered mesenchymal condensations and mechanical cues for tissue regeneration. *Sci Transl Med*. 2019;11.
13. Uyama H, Tu HY, Sugita S, Yamasaki S, Kurimoto Y, Matsuyama T, Shiina T, Watanabe T, Takahashi M and Mandai M. Competency of iPSC-derived retinas in MHC-mismatched transplantation in non-human primates. *Stem Cell Reports*. 2022;17:2392-2408.
14. Lin YT, Liu HL, Day YJ, Chang CC, Hsu PH and Chen JC. Activation of NPFFR2 leads to hyperalgesia through the spinal inflammatory mediator CGRP in mice. *Experimental neurology*. 2017;291:62-73.
15. Yang X, Li Z, Xiao H, Wang N, Li Y, Xu X, Chen Z, Tan H and Li J. A Universal and Ultrastable Mineralization Coating Bioinspired from Biofilms. *Advanced Functional Materials*. 2018;28.
16. Praveschotinunt P, Duraj-Thatte AM, Gelfat I, Bahl F, Chou DB and Joshi NS. Engineered *E. coli* Nissle 1917 for the delivery of matrix-tethered therapeutic domains to the gut. *Nature Communications*. 2019;10.

REVIEWERS' COMMENTS:

Reviewer #1 (Remarks to the Author):

The author has completed the relevant comments submitted by the reviewers, and it is recommended to accept for publication in the current form.

Reviewer #2 (Remarks to the Author):

The authors of the manuscript #COMMSBIO-22-1603A, describing the role of neuropeptide NPVF in bone-brain-interaction and bone regeneration, addressed previous comments and concerns during their revision process accordingly and adjusted their manuscript adequately. They clarified open questions regarding methodology, straightened some descriptions and conclusions, and added suggested datasets about the stability of their material.

Reviewer #3 (Remarks to the Author):

Thank you very much for the careful revision.